# Understanding solvent effects on adsorption and protonation in porous catalysts

Nicholas S. Gould[1], Sha Li[1], Hong Je Cho[1], Harrison Landfield[1], Stavros Caratzoulas[1], Dionisios Vlachos [1], Peng Bai [2✉] & Bingjun Xu [1✉]

Solvent selection is a pressing challenge in developing efficient and selective liquid phase catalytic processes, as predictive understanding of the solvent effect remains lacking. In this work, an attenuated total reflection infrared spectroscopy technique is developed to quantitatively measure adsorption isotherms on porous materials in solvent and decouple the thermodynamic contributions of van der Waals interactions within zeolite pore walls from those of pore-phase proton transfer. While both the pore diameter and the solvent identity dramatically impact the confinement (adsorption) step, the solvent identity plays a dominant role in proton-transfer. Combined computational and experimental investigations show increasingly favorable pore-phase proton transfer to pyridine in the order: water < acetonitrile < 1,4 – dioxane. Equilibrium methods unaffected by mass transfer limitations are outlined for quantitatively estimating fundamental thermodynamic values using statistical thermodynamics.

[1] Catalysis Center for Energy Innovation, Department of Chemical and Biomolecular Engineering, University of Delaware, 150 Academy Street, Newark, DE 19716, USA. [2] Department of Chemical Engineering, University of Massachusetts Amherst, 686 North Pleasant Street, Amherst, MA 01003, USA. ✉email: pengbai@umass.edu; bxu@udel.edu

L iquid-phase reactions mediated by solid catalysts are ubiquitous in chemical synthesis. For example, the low volatility of biomass-derived sugars and furanics requires that a significant fraction of biomass upgrading reactions are conducted in the liquid phase with a solid catalyst, and in many cases in the presence of a solvent[1–5]. Thus, chemical transformations occur at a solid–liquid interface, where the adsorbate energetics are influenced by interaction with the solvent. Even for a solvent that is generally considered as inert, i.e., not directly involved in the molecular transformations, solvation impacts the energetics of adsorbates, intermediates, and transition states (TSs), which could in turn lead to modified rates and selectivities[6–10]. The solvent identity may also result in more direct effects including modifying the properties or oxidation states of catalytic sites or affecting the surface coverage of catalytic sites via competitive adsorption[5,11]. Solvent effects are typically determined through catalytic activity evaluations of specific reactions of interest by correlating a desired performance metric, e.g., product yield/selectivity, with a solvent property, e.g., polarity, basicity, or a substrate's solubility[4,8,12]. However, this approach rarely leads to predictive and generalizable understanding that can guide solvent selection for future reactions of interest. The improved accuracy of computational techniques could allow for deconvoluting solvent effects. While recent works on solvent effects in homogeneous acid catalysis are becoming increasingly quantitative and insightful[13,14], more complex and industrially relevant systems, such as reactions occurring in the micropores of zeolites, are still challenging to model with co-adsorbed solvent. Experimental techniques are needed to generate thermodynamic data that will enable the development of predictive models for such complex reaction systems.

Gas-phase reactions occurring in porous catalysts typically involve an initial adsorption into the pore phase (confinement), and a subsequent interaction with the acid site (either hydrogen-bonded or protonated), before the intrinsic activation step of the chemical reaction. The thermodynamics of liquid-phase reactions are more complex because both substrate and acid site energetics are affected by solvent interactions (Fig. 1)[5,8]. The solvent identity affects the chemical potential of a proton through the formation of a solvent-dependent specific acid ($\mathbf{a} \rightarrow \mathbf{b}$). In the case of a water-saturated zeolite pore, the proton spontaneously transfers to form a charged water cluster (Fig. 1, brown). This water cluster is a specific acid with unique proton transfer properties, compared to the framework-bound proton, which can have a significant impact on catalysis[6,9,14]. The solvent identity also impacts the confinement step ($\mathbf{b} \rightarrow \mathbf{c}$) through interactions that stabilize (or destabilize) the substrate (R) in the external liquid phase ($\Delta H_{solv, B}$) relative to the substrate in the pore phase ($\Delta H_{solv, C}$). Similarly, the hydrogen-bonding/protonation step ($\mathbf{c} \rightarrow \mathbf{d}$) can be affected by the solvent identity because the structure of the specific acid is disrupted upon proton transfer. In this step, a solvent differentially affects the stability of the hydrogen-bonded or protonated adduct ($\Delta H_{solv, D}$) relative to the solvated proton and the pore-phase substrate ($\Delta H_{solv, C}$). The propensity for the proton to transfer to the substrate is affected by solvent interactions with all three of these species and is a major topic of investigation in this work. The solvent may also affect the intrinsic reaction barrier ($E_{a,int}$, step $\mathbf{d} \rightarrow$ TS) by stabilizing the transition state ($\Delta H_{solv, TS}$) relative to the reactant adduct in step $\mathbf{d}$. While Fig. 1 is expressed in terms of enthalpies to emphasize activation barriers (which are purely enthalpic) in the discussion, a similar scheme can also be drawn in terms of Gibbs free energies, as solvent interactions can also have entropic effects that modify a reaction's pre-exponential factor. Experimentally, the effect of solvent on each of the steps in Fig. 1 is challenging to decouple. As a result, one typically resorts to screening many

solvents for the liquid-phase reaction of interest. Yet, identifying the optimal solvent by this brute force approach is time consuming and costly, and often a mixture of several solvents of a specific composition provides the most desired results[2–4,6,15–17]. To elucidate the nuanced roles of solvents and improve the ability to predict solvent behavior, quantitative determination of the impact of solvents on the interactions between substrates and catalytic sites is essential, especially in non-ideal environments such as micropores of zeolites. Further, these data could be used to inform and assess predictive models based on empirical or first principles calculations, eventually leading to model-based selection of solvents. However, quantitative experimental determination of energetics of liquid-phase reactions catalyzed by porous solids is challenging, as slow mass transport in the liquid-phase complicates the application of quantitative methods routinely used for gas-phase reactions, e.g., calorimetry[18].

In this work, attenuated total reflection (ATR)-FTIR is used to measure liquid-phase pyridine adsorption isotherms in zeolites with the goal of quantitatively probing the effect of co-adsorbed solvents on the thermodynamic properties of the pyridine-Brønsted acid site (BAS) interactions. Pyridine is used due to its ability to differentiate Brønsted from molecular adsorption sites (MASs) and its well-known role in characterizing acidity[19–22]. Contributions of pyridine's van der Waals interactions with pore walls are deconvoluted from the thermodynamics of pore-phase proton transfer by comparing the adsorption isotherms on siliceous and aluminosilicate Beta zeolites. For the solvents studied in this work, the free energy of pyridine protonation becomes more negative with decreasing solvent polarity, or dielectric constant. Computational investigations of the preferred structures of solvent-dependent specific acids are used to estimate the pore-phase proton transfer energy (PTE) to pyridine using density functional theory (DFT). Both DFT calculations and experimental adsorption isotherms confirm increasingly favorable PTE in the order: water < acetonitrile < 1,4 – dioxane. The effect of a variety of zeolite properties on pyridine adsorption in water are surveyed in this work, including the zeolite framework structure, silicon to aluminum ratio (Si/Al), and hydrophobicity. In addition to PTE measurements, methods are outlined to estimate several thermodynamic properties of the pore-phase relevant to catalysis in zeolites. These include the change in solvation free energy ($\Delta G_{ads}$), enthalpy ($\Delta H_{ads}$), and entropy ($\Delta S_{ads}$) between the bulk solvent and the pore-phase, as well as pore-phase standard-state chemical potentials and activity coefficients in the presence of co-adsorbed solvents.

## Results and discussion
**Impact of pore size on the adsorption of pyridine.** In liquid water, the pore-phase pyridine concentration sharply decreases in samples with increasing pore diameters from 7 to 15 Å in siliceous microporous materials, and the confinement effect is minimal in a 25 Å pore Si/MCM-41. ATR-FTIR spectroscopy is used to measure pyridine adsorption isotherms in a variety of zeolites and mesoporous materials in liquid water (Fig. 2a). At a constant bulk liquid-phase concentration of pyridine, the equilibrium concentration of pyridine in the siliceous materials is shown to dramatically decrease with increasing pore diameter. At 0.1 M pyridine in the bulk solution (Fig. 2a), the pyridine concentration is only slightly higher (by ~15%) in the MFI framework, which contains 5.2 × 5.7 Å straight channels and 5.3 × 5.6 Å sinusoidal channels, than in the BEA framework which contains a majority of 7.6 × 6.4 Å channels and a minority of 5.5 × 5.5 Å channels[23]. The pore-phase pyridine concentration nears the value for pure liquid pyridine (≈12.4 M) at the concentrated end of the Si/ZSM-5 isotherm, meaning that in Si/ZSM-5, the pores are almost

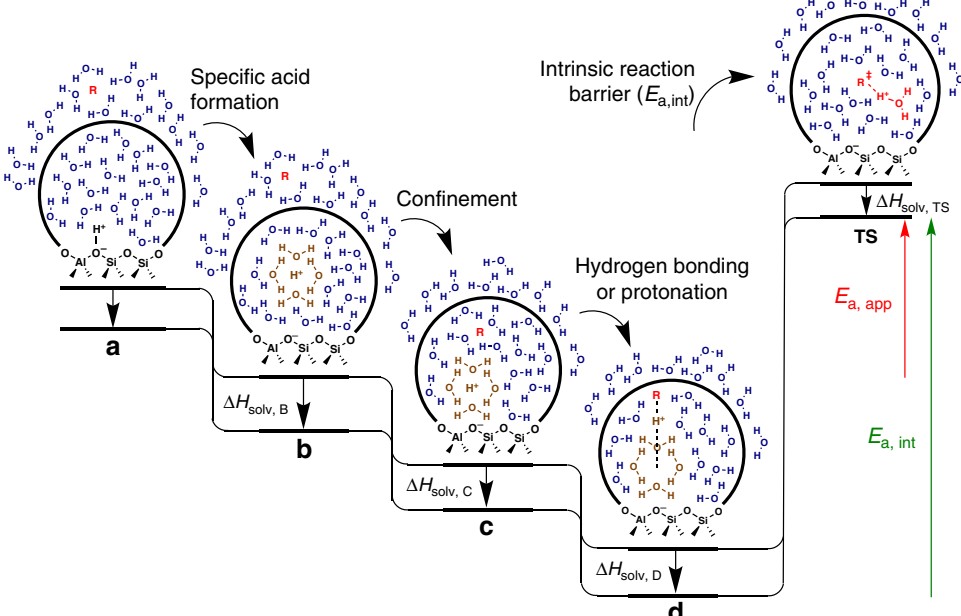

**Fig. 1 Reaction coordinate diagram for a typical zeolite catalyzed, aqueous-phase reaction.** Initially, the reactant (R) is solvated by the liquid phase outside the porous catalyst and the proton in the catalyst is not hydrated (**a**). The hydration of the proton by water forms the specific acid (**b**), which is followed by the adsorption of R in the catalyst pore (**c**). The hydrogen bonding or protonation of R in the catalyst pore is accompanied by a partial or complete de-solvation of the proton (**d**), which is followed by the acid catalyzed chemical transformation by going through a transition state (TS). Water molecules in brown and red are involved in the specific acid solvation sphere and in the TS, respectively. The solid curved lines represent the zeolite pore walls. The enthalpy changes ($\Delta H$), apparent activation energy ($E_{a, app}$), and intrinsic activation energy ($E_{a, int}$) are dependent on interactions with the solvent environment.

exclusively saturated with pyridine with as low as 0.1 M in the feed in liquid water. However, the equilibrium pyridine concentration drops dramatically from Si/Beta to H/USY, which contains 11.8 Å cavities (supercages) where most adsorbed pyridine resides[24], connected by 7.4 Å pore openings. Pyridine has a molecular diameter of roughly 6 Å, and its precipitous drop in pore-phase concentration for samples with pore diameters greater than the roughly 7 Å in the BEA framework is attributed to the loss of the van der Waals stabilization with the pore walls on all sides of the molecule. Three siliceous, ordered mesoporous samples, i.e., Si/MCM-41 (1.5 nm), Si/MCM-41 (2.5 nm), and Si/SBA-15 (6.5 nm) are also synthesized with average pore diameters indicated in the parentheses of Fig. 2a. The pore-phase pyridine concentration continues to decrease dramatically in the 12 Å (H/USY) to 25 Å range, while there is negligible difference between mesoporous materials with the pore diameter ranging from 2.5 to 6.5 nm (Fig. 2b). As the pore diameter increases in the 2.5–6.5 nm range, pyridine molecules only lose marginal contact area with pore walls, which start to resemble a flat surface. As the pore diameter decreases to that of zeolites ZSM-5 and Beta, the adsorption isotherms in Fig. 2 increasingly resemble Langmuir isotherms, as pore-phase pyridine molecules experience a relatively strong interaction with pore walls. As the diameter increases to the extent that a second or more "layers" are sterically possible (Fig. 2c), pore-phase pyridine molecules can exist in multiple distinct configurations and interact predominantly with neighboring pyridine or solvent molecules at the expense of direct contact with pore walls (the equivalent of the surface in the Langmuir model). The increased adsorbate-surface interaction strength for Beta and ZSM-5 relative to the adsorbate-adsorbate and the adsorbate-solvent interactions for mesoporous materials results in isotherms with greater Langmuir character. Both the identical adsorption site and the non-interacting adsorbate assumptions are more valid for pore diameters of similar

dimensions to the adsorbate. Further discussion and corroboration of the applicability of the Langmuir model to ZSM-5 and Beta isotherms in water can be found in the Supplementary Discussion section surrounding Supplementary Eqs. 1, 2 and Supplementary Table 1. Note that several thermodynamic properties can be estimated from the isotherms in Fig. 2a. The free energy of transfer ($\Delta G_{ads}$) for the adsorption isotherms in Fig. 2a are calculated using the relative concentrations of pyridine in the external liquid (L) and the pore phase (Z) using Eq. 1 at constant temperature and pressure (T and P, Supplementary Fig. 1, and the Methods section). Note that Eq. 1 and all thermodynamic properties computed in this work are not dependent on the use of the Langmuir model. For example, the adsorption free energy can be calculated for each data point along the adsorption isotherm via Eq. 1. Additional fundamental thermodynamic properties of the pore phase computed from Fig. 2a include the pore-phase pyridine standard-state chemical potentials ($\mu_Z^\circ$) and pore-phase activity coefficients ($\gamma_Z$), which are listed in Supplementary Table 2 and Supplementary Fig. 2, respectively. Note that all pore-phase concentrations and thermodynamic properties computed in this work are ensemble averages, and zeolite pores do consist of multiple adsorption sites. However, this situation is not unlike that of a microheterogeneous solution, where aggregation and molecular clustering lead to variations in local concentration and modify the environment surrounding any particular solute species[14]. This is also the quantity conventionally used in the analysis of chemical kinetics[11], even for catalysts exhibiting high microheterogeneity, i.e., a total surface concentration of CO on supported metal catalysts with a diversity of adsorption sites. A full analysis and discussion of the thermodynamic properties estimated in this work can be found in the Methods and Supplementary Discussion sections, alongside additional pyridine adsorption isotherms for several H/ZSM-5 and H/Beta samples with varying Si/Al ratios and hydrophilic/hydrophobic textures

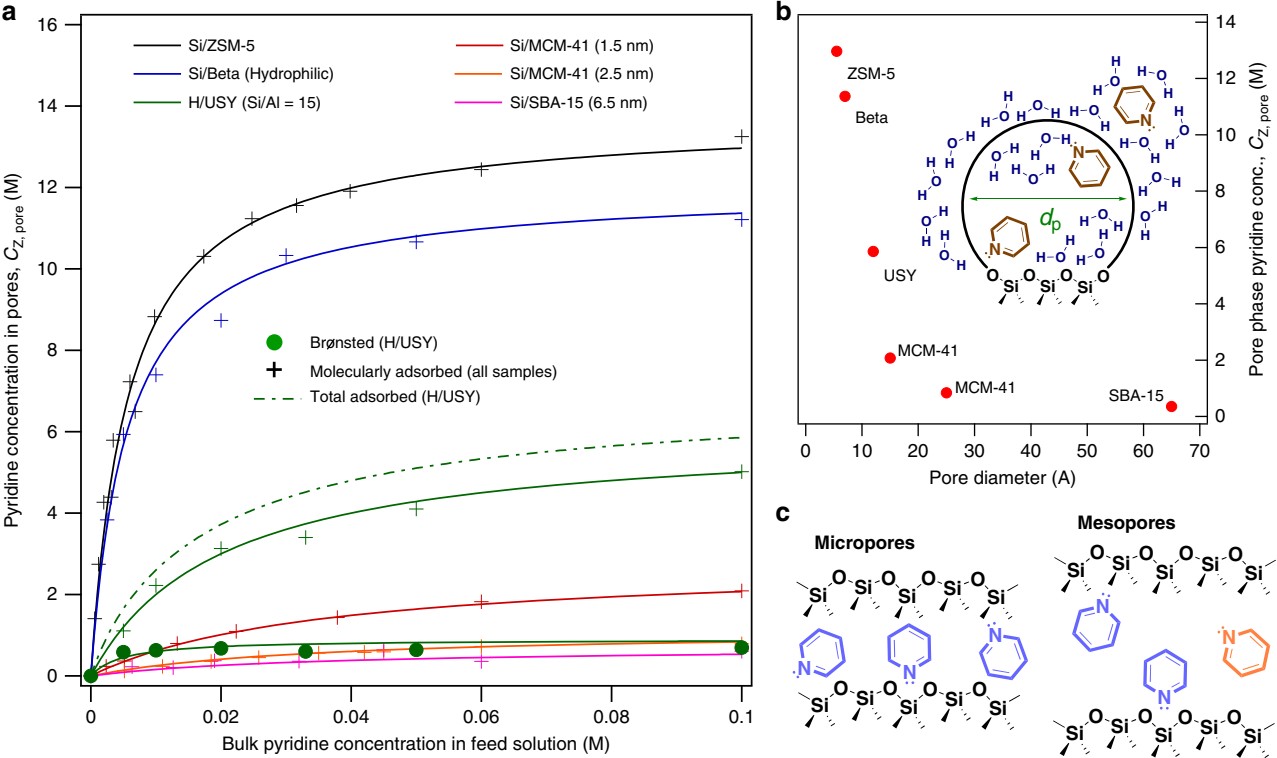

**Fig. 2 Adsorption of pyridine on siliceous porous materials in liquid water. a** Pyridine adsorption isotherms on adsorbents of varying average pore diameters in liquid water in ATR-FTIR at 20 °C. The pore-phase pyridine concentration on MAS is denoted by "+" signs and the pyridinium is denoted by dots. The sum of the pyridine on BAS and MAS is denoted with a dashed line for H/USY. **b** The pore-phase pyridine concentration with 0.1 M pyridine in the external solvent as a function of pore diameter. The ATR-FTIR spectra for the data in Fig. 2 can be found in Supplementary Figs. 26-31. Note that aluminosilicate FAU is used because siliceous FAU cannot be synthesized. Inset: depiction of the adsorbent pore diameter ($d_p$). **c** Depiction of pyridine adsorption in micropores and mesopores where molecules of identical color experience similar adsorption energetics.

(Supplementary Fig. 3).

$$\Delta G_{\text{ads}} = -RT \ln\left(\frac{C_Z}{C_L}\right) \quad (1)$$

**Impact of solvents on the adsorption of pyridine.** The concentration of pyridine in hydrophilic Si/Beta pores decreases in the order water > acetonitrile > ethanol > 1,4 – dioxane, which can likely be rationalized based on liquid-phase pyridine activity coefficients and the pore-phase stability of solvent molecules (Fig. 3a). The adsorption isotherm on Si/Beta is most dramatic in the case of water, which has a Langmuir adsorption constant ($K'$) equal to $207 \pm 9$ (Fig. 3a legend). This experiment is repeated multiple times (Supplementary Fig. 4), resulting in a standard error of the mean equal to 9. In a previous work, pyridine and pyridinium extinction coefficients on zeolites in solvent were shown to have standard deviations ranging from 8 to 18% depending on the solvent of interest[25]. The $K'$ value drops by about one order of magnitude for acetonitrile. In the cases of ethanol and 1,4-dioxane, the isotherm is approximately linear from 0 to 0.1 M (still within Henry's regime). The energetic properties of the Si/Beta pore phase in the four solvents in Fig. 3a can be quantified and fully described by standard-state chemical potentials ($\mu^\circ_{\text{pyr,Z}} - \mu^\circ_{\text{pyr,L}}$) and pore-phase activity coefficients (Table 1 and Supplementary Fig. 5, respectively). In Table 1, the columns describe the differences in standard-state chemical potentials between all combinations of three phases: zeolite, liquid, and an ideal gas vapor phase (Fig. 3b). Note that as an ideal gas, $\mu^\circ_{\text{pyr,V}}$ is solvent independent. Thus, the solvent dependence in $\mu^\circ_{\text{pyr,L}} - \mu^\circ_{\text{pyr,V}}$ in Table 1 is entirely based on dilute

**Table 1 Henry's Law standard-state chemical potential differences in Si/Beta at 20 °C and 1 bar.**

| Solvent | $\mu^\circ_{\text{pyr,Z}} - \mu^\circ_{\text{pyr,L}}$ (kJ mol$^{-1}$) | $\mu^\circ_{\text{pyr,L}} - \mu^\circ_{\text{pyr,V}}$ (kJ mol$^{-1}$) | $\mu^\circ_{\text{pyr,Z}} - \mu^\circ_{\text{pyr,V}}$ (kJ mol$^{-1}$) |
|---|---|---|---|
| Water | −15.21 | −13.73 | −28.94 |
| Acetonitrile | −9.05 | −16.89 | −25.94 |
| Ethanol | −4.48 | −16.63 | −21.11 |
| 1,4 – Dioxane | −2.39 | −15.50 | −17.89 |

$\mu^\circ_{\text{pyr,Z}}$ is 1 M pore-phase pyridine based on extrapolation of dilute solution behavior.
$\mu^\circ_{\text{pyr,L}}$ is 1 M liquid pyridine based on extrapolation of dilute solution behavior.
$\mu^\circ_{\text{pyr,V}}$ is pure, ideal pyridine gas at 1 bar.
Calculated values include the volume occupied by the zeolite framework.

pyridine interactions with the corresponding liquid solvent. These values are estimated using $P$-$x$-$y$ diagrams in Aspen (see Methods section). The difference between the zeolite and vapor-phase chemical potentials is listed in the right-most column and is the sum of zeolite-liquid and the liquid-vapor columns, which can be directly compared across different solvents (by cutting out the "middleman" liquid phase). The importance of decomposing the adsorption from liquid phase via a thermodynamic Haber cycle and comparing these three columns is that one can quantitatively estimate how an adsorption isotherm (Fig. 3a) is influenced by both liquid-phase and zeolite-phase interactions, i.e., viewing liquid-phase adsorption into pores as a pathway involving desolvation followed by vapor-phase adsorption (Fig. 3b). In the bulk liquid, the effect of solvent interactions on pyridine's free energy is quantified via infinite-dilution activity coefficients based on Raoult's law (Fig. 3a legend, subscript R). Note that

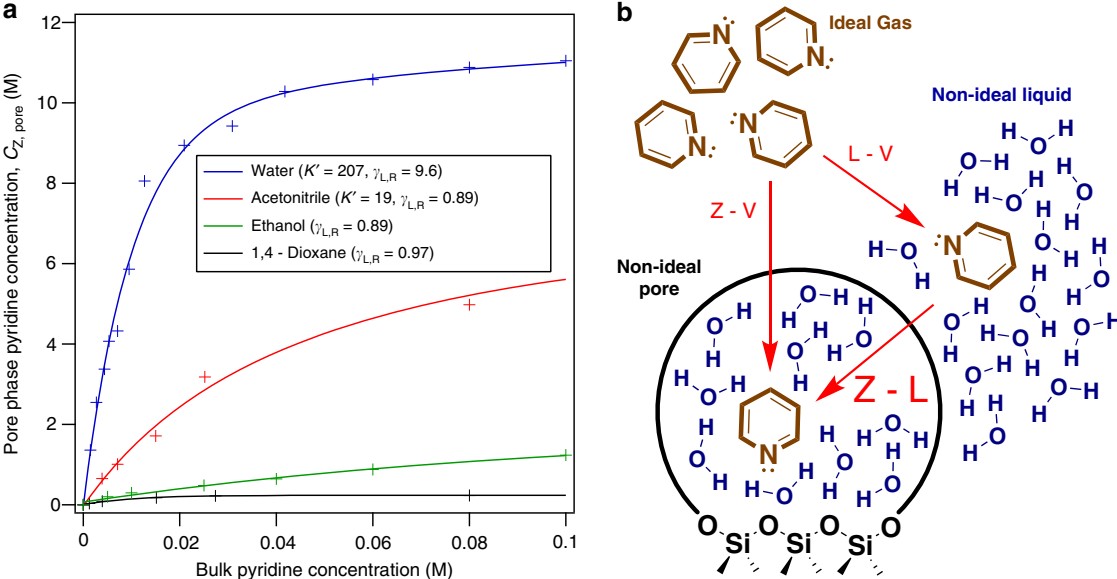

**Fig. 3 Pyridine adsorption isotherms on siliceous Beta in various solvents. a** Pyridine adsorption isotherms on Si/Beta (hydrophilic) at 20 °C in various solvents. **b** A Haber cycle for vapor-liquid-zeolite pyridine adsorption into water-saturated Si/Beta. Ideal gas pyridine may be used as a common reference state for all solvents to deconvolute the energetic contributions of bulk liquid solvation outside the pore from interactions within the pore. Z, L, and V refer to the zeolite, liquid, and vapor phases, respectively.

pore-phase activity coefficients in the Methods section are defined by the Henry's law definition (subscript H). The Raoult's definition is a convenient choice for comparing the relative free energies of interaction with the solvent environment in the bulk liquid-phase because in all solvents, Raoult's activity coefficients are defined from a common reference state: pure liquid pyridine at 1 bar. This is not true of Henry's law activity coefficients, whereas one changes the identity of the solvent, the reference state is infinitely dilute pyridine in the particular solvent of interest, making energetic comparisons across solvents non-trivial. In water, the higher liquid-phase activity coefficient (Raoult's) compared to the non-aqueous solvents is indicative of a less favorable free energy of transfer from the ideal vapor phase into the non-ideal solvent environment. The higher activity coefficient in water is a combination of a higher enthalpy from interactions of the mostly hydrophobic pyridine molecule with water, as well as the disruption of water's hydrogen-bonding network. These factors can contribute to a higher pore-phase concentration. However, liquid-phase activity coefficients will never be entirely predictive of the pore-phase concentration, as they are only representative of the energetics of one of the two phases (the external liquid, but not the pore). The similar liquid-phase activity coefficients ($\gamma_{L,R}$) for the three organic solvents indicates that pyridine has similarly stable interactions with the external solution in these cases, and the stark differences in the adsorption isotherms must be dictated by differences within the pore phase (Fig. 3a). For example, the activity coefficients in the legend of Fig. 3a suggest that pyridine is less stable in liquid water than in the organic solvents (this is reflected in the L–V column in Table 1). However, it is unclear from activity coefficients to what extent pyridine's affinity for the pore in water is due to liquid-phase interactions versus pore-phase interactions. This comparison is made quantitative using the Haber cycle values in Table 1. As an example, when comparing the free energy of pyridine adsorption into Si/Beta in water to acetonitrile, the Z–L column shows a net difference of roughly 6.2 kJ mol$^{-1}$, while the Z–V column is about 3.0 kJ mol$^{-1}$ apart. Because the vapor phase is solvent independent (ideal gas), the differences between the various solvents in the Z–V column are entirely due to

pore-phase interactions (which include interactions both with pore walls and with adsorbed solvent molecules, Fig. 3b). Thus, by comparing the Z–L and the Z–V columns for water and acetonitrile, roughly 3.2 kJ mol$^{-1}$ out of the 6.2 kJ mol$^{-1}$ difference in Z–L adsorption is due to pyridine's unfavorable interactions with the external bulk liquid water, while the other 3.0 kJ mol$^{-1}$ is due to favorable interactions in the zeolitic phase in the presence of water. Thus, this Haber cycle framework is an ideal way to reduce the amount of guesswork and qualitative statements involved in predicting pore-phase adsorption properties. Extending this analysis to the non-aqueous solvent isotherms in Fig. 3a, the differences in pore-phase pyridine concentration is almost entirely due to pore-phase differences (opposed to bulk solvent interactions) and could be the result of competitive adsorption (confinement) of the solvent. At room temperature, this competition is mostly dependent on enthalpic contributions. As an example, 1,4 – dioxane, as well as pyridine, have a molecular size comparable to the size of the micropores in Si/Beta, which could result in similarly strong van der Waals contacts with zeolite pore walls. Due to their smaller molecular size, ethanol and acetonitrile likely have less pore-phase enthalpic stabilization compared to 1,4-dioxane, but ethanol's affinity for zeolite pores has been noted in the literature[26–28]. A plot of the pore-phase activity coefficients in Si/Beta based on the isotherms in Fig. 3a is included in Supplementary Fig. 5. Alongside the standard-state chemical potentials in Table 1, pore-phase activity coefficient values allow one to map the chemical potential of the pore-phase pyridine (or a reaction substrate) across the entire concentration range.

**Deconvolution of confinement and protonation.** For the solvents studied in this work, saturation of BAS by pyridine becomes more favorable with decreasing solvent polarity. Pyridinium peak areas on H/Y (Si/Al = 3) and H/Beta (Si/Al = 12) are tracked with increasing liquid pyridine concentration in four solvents of varying dielectric constants ($\varepsilon$, Fig. 4, spectra in Supplementary Figs. 6–12). The isotherms are normalized according to the saturation loading on BAS to highlight the slight change in the curve shape of the isotherms. Note that $K'$ increases as the solvent

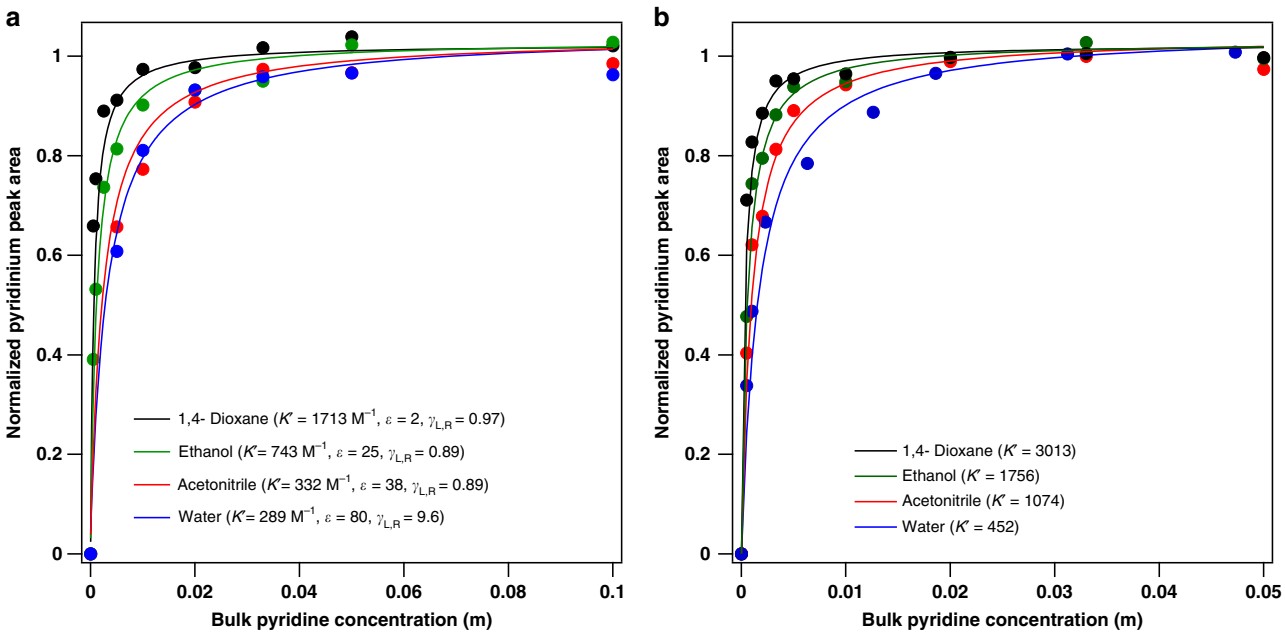

**Fig. 4 Normalized pyridine isotherms in various solvents on aluminosilicate zeolites. a** Pyridine adsorption isotherms onto H/Y BAS (Si/Al = 3) and **b** H/Beta (Si/Al = 12) in various solvents at 20 °C. Dielectric constants ($\varepsilon$), activity coefficients ($\gamma$), and Langmuir parameters ($K'$) listed in legend.

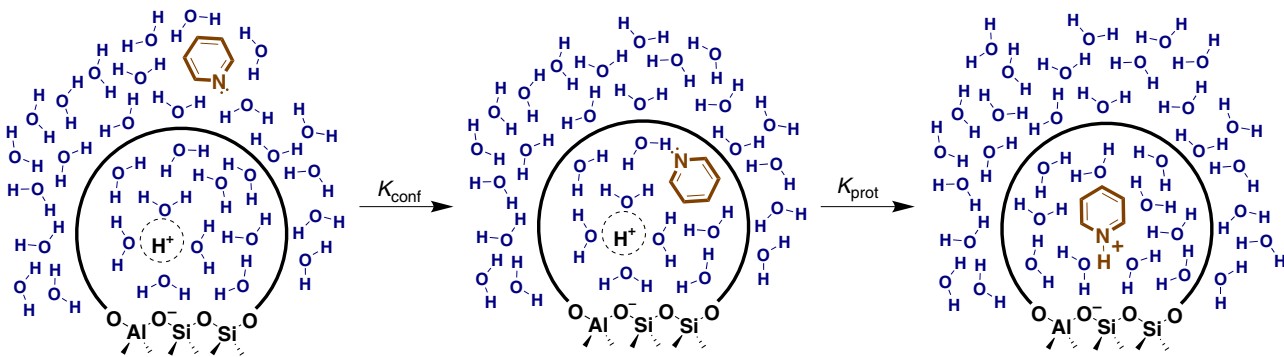

**Fig. 5 Depiction of the confinement ($K_{conf}$) and the protonation ($K_{prot}$) thermodynamic steps.** Pyridine enters the catalyst pore with an equilibrium constant of $K_{conf}$ before being protonated by the BAS with an equilibrium constant of $K_{prot}$. Solid and dashed curved lines indicate zeolite pores walls and the BAS solvation sphere, respectively.

polarity decreases, from water to 1,4 – dioxane, although the trend is not linear. However, isolating the information regarding the acid strength from isotherms and equilibrium constants derived from Fig. 4 is difficult, as any thermodynamic parameters determined from these isotherms represent the external liquid pyridine both entering the pore ($K_{conf}$, Eq. 2) and subsequently getting protonated ($K_{prot}$, Fig. 5 and Eq. 3), where L and Z denote the liquid phase and the zeolite pore phase, respectively. The confinement equilibrium constant ($K_{conf}$) involves liquid-phase pyridine entering the zeolite pore and displacing pore-phase solvent into the bulk liquid-phase (Eq. 4), while the protonation equilibrium constant ($K_{prot}$) reflects proton transfer from the solvent-dependent specific acid to pore-phase pyridine (Eq. 5). The stability of protons in these specific acid solvent-clusters will be further discussed alongside DFT calculations later in this work. In the vapor phase, van der Waals interactions are known to affect heats of adsorption of probe molecules onto BAS (as an acidity measurement)[29–31]. Similarly, in the liquid phase, it is apparent from the adsorption isotherms on Si/Beta (Fig. 3a) that the solvent choice impacts the thermodynamics of confinement. The adsorption isotherms into purely siliceous zeolites can

provide an estimate of the dispersion forces ($K_{conf}$) to remove the effect from apparent acidity measurements.

$$\text{Pyridine}_L + \text{Solvent}_Z \leftrightarrow \text{Pyridine}_Z + \text{Solvent}_L \quad \text{(confinement)} \quad (2)$$

$$\text{Pyridine}_Z + \text{SolventH}_Z^+ \leftrightarrow \text{PyridineH}_Z^+ + \text{Solvent}_Z \quad \text{(proton transfer)} \quad (3)$$

$$K_{conf} = \frac{\left(C_{pyr,Z} \times C_{solvent,L}\right)}{\left(C_{pyr,L} \times C_{solvent,Z}\right)} \quad (4)$$

$$K_{prot} = \frac{\left(C_{pyrH^+,Z} \times C_{solvent,Z}\right)}{\left(C_{pyr,Z} \times C_{solventH^+,Z}\right)} \quad (5)$$

The thermodynamics of proton transfer are significantly more sensitive to the solvent choice than to the zeolite framework. The confinement and subsequent protonation of pyridine depicted in Fig. 5 is modelled as a reaction in Eqs. 6 and 7, which includes both

**Table 2 Confinement and proton transfer equilibrium constants near infinite dilution in solvent.**

| Solvent (Zeolite) | $K_{conf} K_{prot}$ | $K_{conf}$ | $K_{prot}$ | $\varepsilon$ |
|---|---|---|---|---|
| 1,4 – Dioxane (H/Beta) | 37500 | 9.1 | 4120 | 2 |
| Ethanol (H/Beta) | 21800 | 31 | 703 | 25 |
| Acetonitrile (H/Beta) | 13400 | 158 | 85 | 38 |
| Water (H/Beta) | 5610 | 1980 | 2.8 | 80 |
| Water (H/Y) | 3580 | 575 | 6.3 | 80 |

the confinement and the proton transfer steps (Eq. 7 is the product of Eqs. 4 and 5). This combined reaction must be used when applying any values taken from Fig. 4, because both confinement and proton transfer from the specific acid apply to the isotherms on BAS. All of the variables in Eq. 7 can be obtained from Fig. 4, including the concentrations of pyridinium and the protonated solvent clusters, which can be estimated over the entire concentration range as the saturated concentration of pyridinium ($C_{pyrH^+,max}$) minus the pyridinium concentration ($C_{pyrH^+}$) at a given bulk liquid pyridine concentration ($C_{pyr,L}$). The combined $K_{conf} K_{prot}$ equilibrium constant near infinite dilution ($5 \times 10^{-4}$ M) decreases with increasing polarity, or $\varepsilon$, of the solvent (Table 2). The equilibrium constant due to confinement ($K_{conf}$) is estimated in Table 2 using the relative pore-phase and liquid-phase pyridine concentrations at infinite dilution in purely siliceous Si/Beta in Fig. 3a. From the Si/Beta isotherms, $K_{conf}$ increases in the order: 1,4-dioxane < ethanol < acetonitrile < water. For the H/Y sample in water (Table 2), the $K_{conf}$ value is estimated using the H/USY (Si/Al = 15) MAS isotherm from Fig. 2a near infinite dilution. The greater $K_{conf}$ value for BEA than for USY zeolites is consistent with the increased pyridine uptake in BEA as discussed in the context of the effect of the pore diameter on pyridine uptake (Fig. 2a). The equilibrium constant for the proton transfer step ($K_{prot}$) is estimated by removing the $K_{conf}$ term from the combined $K_{conf} K_{prot}$ values in Table 2. The trend in $K_{prot}$ follows the same trend as the combined $K_{conf} K_{prot}$ quantity, increasing for decreasingly polar solvents from water to 1,4-dioxane. The $K_{prot}$ values are closely related to the apparent acidity of the BAS (as it exists in a protonated solvent cluster). However, acidity is often defined as the free energy of the acid dissociation (deprotonation) reaction (GDPE) of the neutral acid[32]. The use of a proton transfer reaction to a base like pyridine is an experimental surrogate measurement of acidity that does introduce complications, several of which are unique to the liquid phase[33]. In solvent, the proton is often detached from the negatively charged zeolite framework (conjugate base) and exists in a charged cluster of solvent molecules (the specific acid). This idea is depicted for water in Fig. 5, where the proton is delocalized among a cluster of a small number of water molecules[34–39]. Note that the identity of the solvent will have a significant impact on the structure and stability of the proton and thus its ease of transfer to a base such as pyridine[40]. One additional complexity unique to the liquid phase is the possibility that the preferred structure of the solvent-proton cluster is affected by the location of the BAS in the zeolite pore, as steric limitations could cause bulkier solvent molecules to form clusters of different structures and thus different local acidities. Further, the solvent identity may also affect the energetics of the pyridinium interactions with the negatively charged zeolite framework, a topic that has been discussed as affecting vapor-phase acidity measurements[31]. Note that $K_{conf}$ only accounts for the non-ionic component of the interactions pyridinium experiences with the pore-phase solvent environment. It is likely that the solvent would have some impact in how charge in both pyridinium and the zeolite framework are distributed and stabilized. Thus, the $K_{prot}$ term is an aggregate of

ionic effects: the stability of the proton, pyridinium, and the framework charge both before and after proton transfer. After accounting for the confinement effect (van der Waals stabilization), the protonation of pyridine is still more facile in the low polarity solvents. Most apparent is the drop in $K_{prot}$ from acetonitrile to water, which spans more than one order of magnitude. This suggests that compared to the other solvents, water has a uniquely strong ability to stabilize the proton (BAS) relative to its ability to stabilize pyridinium. Despite the dramatic impact of the solvent identity on $K_{prot}$, the H/Y value is only slightly greater than that of H/Beta. This is in general agreement with periodic DFT calculations which have estimated similar acidities (by DPE) of isolated BAS regardless of the framework in the literature[30]. In homogeneous liquid-phase catalysis by BAS, the effect of the solvent on the relative stability of protons versus protonated transition states has been demonstrated to affect reaction rates and selectivities[14]. This shift in the apparent rate constants is prevalent in work by the Dumesic group, where BAS catalyzed rates of 1,2-propanediol to propanal, and cellobiose to glucose increased in gamma valerolactone (GVL) compared to water due to the relative stabilization of the proton compared to the protonated transition state[6]. This idea is discussed for a general reaction in Fig. 1, where the apparent and intrinsic activation barriers are affected by solvent interactions. More recent work of BAS catalyzed dehydration and hydrolysis reactions studied how the energetics of proton transfer to reactants with varying hydroxyl groups densities (hydrophilicity) is affected by the composition of organic solvents mixed with water. Organic co-solvents caused water-enriched domains to form in the vicinity of reactant hydroxyl groups, increasing hydroxyl group hydrogen-bonding strength with neighboring water molecules, and leading to stabilized proton transfer and protonated transition states compared to pure water[41]. An exhaustive study of mixtures of water with either DMSO, dioxane, THF, GVL, or acetonitrile demonstrated that molecular-scale interactions form multicomponent solvent clusters with highly varying proton affinities compared to either pure component. The heat of mixing of the two solvent components was proposed as a metric to describe the relative proton versus transition state stabilities, and thus the reactivities of a series of reactants of varying hydrophilic character[14]. Concerning experimental methods, in recent work from the Shanks group, high-resolution MAS NMR was used to study mixed solvent interactions with BAS at the solid–liquid interface over supported sulfonic acid materials as well as H/ZSM-5[9]. The technique was able to distinguish bulk water from water interacting with the BAS, and used water's chemical shift as a metric of the relative acidity over the composition range in a mixture with d6-DMSO. Note that for any spectroscopic technique, including the ATR-FTIR method in this work, there are two potential limitations regardless of whether the technique is used for adsorption or for in-situ reaction application. The first is that the technique should be able to distinguish the species of interest, e.g., pyridine vs. pyridinium in this case, or reactant vs. intermediate, and the second is that the species of interest, e.g., intermediates, should be stable enough to be observable. Modular excitation spectroscopy (MES) with phase-sensitive detection (PSD) can improve the surface-sensitivity of ATR-FTIR for adsorbates and surface-intermediates in many cases where these species are challenging to observe[42,43].

$$\text{Pyridine}_L + \text{SolventH}_Z^+ \leftrightarrow \text{PyridineH}_Z^+ + \text{Solvent}_L \qquad (6)$$

$$K_{conf} K_{prot} = \frac{\left( C_{pyrH^+,Z} \times C_{solvent,L} \right)}{\left( C_{pyr,L} \times C_{solventH^+,Z} \right)} \qquad (7)$$

DFT calculations of solvent interactions with the BAS in zeolite pores confirm that the relevant acid complex involved in liquid-phase pyridine protonation in zeolites is not necessarily a bare Al-O-Si BAS, but a protonated solvent cluster (Fig. 1). Namely, in water and acetonitrile, the thermodynamically favored acid site is the specific acid: a hydrated $H(H_2O)_n^+$ cluster in water and a protonated dimer in acetonitrile ($[(CH_3CN)_2H]^+$). This proton transfer to the respective solvent cluster is non-activated and exoergic, as revealed by 2 picosecond (ps) ab initio molecular dynamics (AIMD) simulations and subsequent optimization of low-energy configurations sampled from the trajectories. Full computational details can be found in the Methods section, including a further discussion about the optimization of the number of co-adsorbed solvent molecules. The spontaneous protonation of water to form the protonated water cluster specific acid is highly favorable. As a point of reference, the proton affinity of a water trimer is similar to that of ammonia[44], ca. 9 eV, and the facile formation of hydrated $H_3O^+$ in zeolites has been extensively discussed in the literature[44–48]. A single acetonitrile molecule does not become spontaneously protonated by the Al-O-Si bridge BAS, while the dimeric $[(CH_3CN)_2H]^+$ cluster is basic enough to spontaneously abstract the proton. The protonation of acetonitrile in zeolites is known to be difficult[49–51], as it only occurs at high acetonitrile pressure, high temperature, or in small pores (e.g., side pockets of mordenite zeolites)[52–57]. Based on FTIR studies in H/ZSM-5, the protonated acetonitrile species—when formed—has been assigned to a dimer[58,59]. Thus, for water and acetonitrile solvents, the PTE to pore-phase pyridine is investigated in separate calculations from two distinct acid sites: from the specific acid site (the protonated solvent cluster), and from the Al-O-Si bridge site. Note that the PTE from the specific acid is equivalent to the pore-phase proton transfer reaction in Eq. 3. The DFT optimized structure of the initial state, which involves neutral pyridine and a protonated water cluster, and the final state with pyridinium and co-adsorbed water molecules are shown in Fig. 6. In the case of 1,4-dioxane, simulations do not exhibit spontaneous proton transfer to the solvent, and thus only proton transfer from the Al-O-Si bridge site is investigated. DFT calculated structures in non-aqueous solvents are included in the Supplementary Figs. 32–36.

The calculated PTEs from the specific acid exhibit the same trend of the solvent effect on apparent acidity as the trend in $K_{prot}$ (Table 3). The PTE is defined as the difference between the energy of the optimized pyridinium + conjugate-base pair and the energy of the optimized pyridine + neutral acid complex. Consider the specific acid in the case of water and acetonitrile, the ease of proton transfer to pyridine increases in the order: water < zeolite in vacuum < acetonitrile < dioxane (Table 3). This trend in PTE in solvent agrees with the ranking order of $K_{prot}$ constants presented in Table 2. The water clusters effectively stabilize the $H_3O^+$ ion, causing proton transfer to pyridine to be roughly 0.24 eV less facile in water than from the bridge site of the zeolite in vacuum. Ignoring the experimentally-confirmed, specific acid solvent clusters in the PTE calculations in water and acetonitrile would result in a dramatically different ranking order of acidity (Supplementary Table 3). Thus, the treatment of the structure and stability of the protonated solvent cluster is critically important in obtaining reliable PTE values.

**Temperature dependence of pyridine adsorption**. The measured enthalpy and entropy of adsorption ($\Delta H_{ads}$ and $\Delta S_{ads}$) of pyridine in Si/ZSM-5 from liquid water are −30 kJ mol$^{-1}$ and −53 J mol$^{-1}$ K$^{-1}$, respectively, which are lower in magnitude than the gas phase equivalent. These values are estimated based on pyridine adsorption isotherms at multiple temperatures

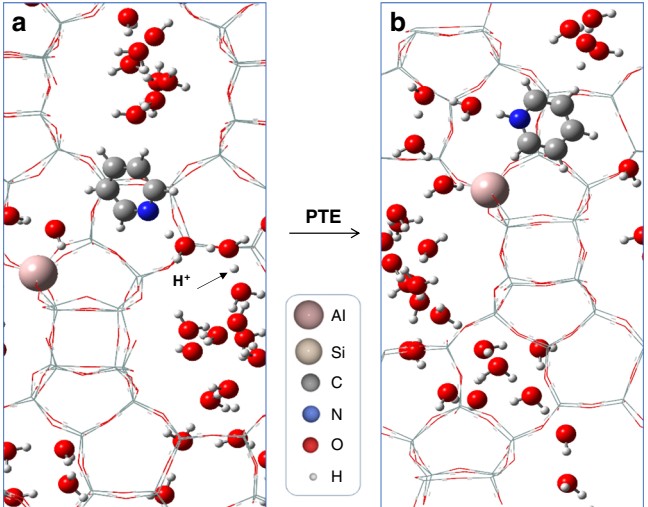

**Fig. 6 DFT Optimized H/Beta structures with 28 water molecules per unit cell demonstrating the PTE. a** The specific acid (protonated water cluster) structure and neutral pyridine and **b** pore-phase pyridinium (PyrH$^+$) and neutral co-adsorbed water.

**Table 3 Proton transfer energy (PTE) to pyridine in H/beta in solvent.**

| System | BAS Structure | PTE (eV) |
|---|---|---|
| H/Beta in Vacuum | Al-OH | −0.67 |
| H/Beta in 1,4-dioxane | Al-OH | −0.85 |
| H/Beta in Acetonitrile (H/Beta) | $H(CH_3CN)_2^+$ | −0.70 |
| H/Beta in Water | $H(H_2O)_n^+$ | −0.43 |

(Fig. 7a, and Supplementary Figs. 13–18) and by plotting $\Delta G_{ads}$ against temperature using the Gibbs energy relationship ($\Delta G = \Delta H - T\Delta S$, Fig. 7b). $\Delta G_{ads}$ values are estimated using the relative concentration of pyridine in each phase near infinite-dilution in Eq. 1 (Methods section and Supplementary Fig. 1). The general linear distribution of the data points indicates that $\Delta H_{ads}$ and $\Delta S_{ads}$ are relatively temperature independent in the temperature range studied (20–80 °C). It is worth noting that these $\Delta H$ and $\Delta S$ values are markedly lower in magnitude than vacuum or gas-phase pyridine adsorption values in zeolites. Gas-phase pyridine adsorption enthalpies are roughly −200 kJ mol$^{-1}$ for adsorption onto H/ZSM-5 (including protonation)[60] and −100 kJ mol$^{-1}$ onto a silica surface (no protonation)[61]. Gas-phase pyridine adsorption into siliceous MFI zeolites that includes confinement, but not protonation, likely falls between these two values. The less extreme adsorption enthalpy in the liquid phase compared to the vapor phase could be attributed to at least two factors. Interactions of the adsorbate with bulk phase solvent typically provide stabilization via intermolecular interactions that are absent in vacuum and are significantly weaker in the gas phase. In this case, the pyridine solvation energy in water is roughly −50 kJ mol$^{-1}$ (referenced to gas phase pyridine), meaning that the liquid-phase pyridine molecule starts in a more stable initial state than vapor phase pyridine[62]. A second potential factor contributing to weakened adsorption enthalpy in solvent is the displacement of previously adsorbed solvent molecules upon adsorption of the substrate. For water, the experimental adsorption enthalpy from the gas phase into a Na/ZSM-5 sample is in the −40 to −50 kJ mol$^{-1}$ range for water molecules not interacting with cations[63], and theoretical calculations for water adsorption onto silicalite equal −52 kJ mol$^{-1}$,[64]. By comparing to water's heat of

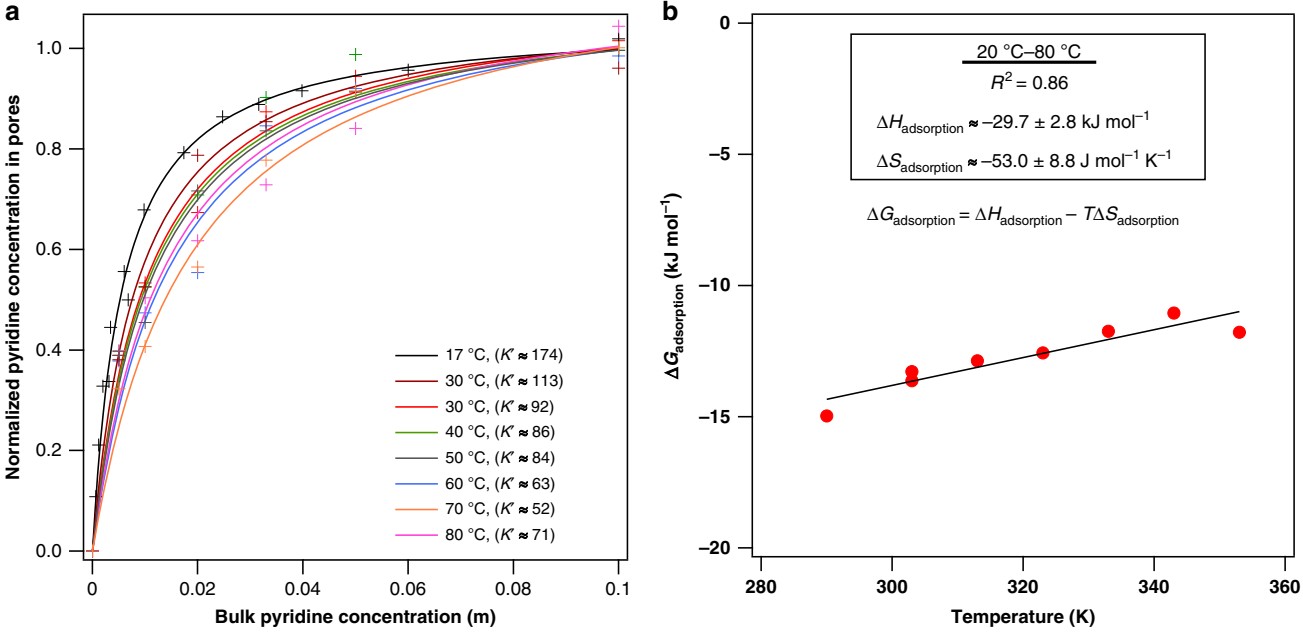

**Fig. 7 Pyridine adsorption isotherms on Si/ZSM-5 at different temperatures. a** Pyridine adsorption isotherms onto Si/ZSM-5 in water at various temperatures. Langmuir parameters ($K'$) in legend. **b** Infinite dilution $\Delta H_{ads}$ and $\Delta S_{ads}$ from 20 to 80 °C in water on Si/ZSM-5. Error bars in B) are standard deviations.

vaporization (41 kJ mol$^{-1}$), this means that water displacement will incur a roughly $-5$ to $-10$ kJ mol$^{-1}$ enthalpic penalty for desorption from pores into the external liquid. The relative molecular sizes of the substrate and solvent molecules is also relevant, as the adsorption of a large substrate molecule could displace more than one adsorbed solvent molecule. Thus, the measured $\Delta H_{ads}$ for pyridine adsorption on Si/ZSM-5 in liquid water is generally consistent with literature values in the gas phase corrected by the solvation enthalpy of pyridine in water and the enthalpic cost of displacing adsorbed water in silicalite. The reduced entropy loss for adsorption from water compared to that from vacuum follows a similar logic, in that molecules in gas phase or vacuum have very few restrictions on translational/rotational/vibrational degrees of freedom, and interactions with bulk solvent would restrict those degrees of freedom. The entropy of solvation of pyridine from the gas phase to water is about $-120$ J mol$^{-1}$ K$^{-1,65}$, while the entropy of adsorption of gas-phase pyridine into H/ZSM-5 is about $-160$ to $-170$ J mol$^{-1}$ K$^{-1,61}$. The $\Delta S_{ads}$ into the pore from the liquid phase in Fig. 7b ($-53$ J mol$^{-1}$ K$^{-1}$) agrees well with these values, and suggests that the main reason for the less extreme entropy loss upon adsorption from the liquid phase compared to the gas phase is due to differences in the external (bulk liquid) phase properties, rather than a unique property of the pore-phase environment. Note that the weakened adsorption entropy loss in the liquid-phase has been observed in the literature, even to the extent where the entropy increases upon confinement[66]. In the vapor phase, adsorption is most commonly an exothermic process and the decrease in translational/rotational degrees of freedom typically leads to a decrease in entropy[67]. In liquid-phase adsorption, an increase in entropy and even endothermicity is more common than in the vapor phase. Examples are prevalent in liquid water, such as the adsorption of ions from aqueous solution where the energy required to disrupt the hydration sphere exceeds the stabilization due to bonding to the solid surface[68,69]. An increase in adsorption entropy from the aqueous phase is common in the adsorption of large-molecular dyes onto solid surfaces, as a large number of solvent molecules

are displaced per mole of adsorbate[68,70]. Positive entropies can also result in the case of adsorption of hydrophilic substrates, e.g., glucose, from aqueous solution. Hydrogen-bonding interactions of hydrophilic functional groups with liquid water can be disrupted upon confinement into microporous materials such as zeolites. This was observed in the case of glucose adsorption from aqueous solution into zeolite Beta[66]. Further examples of endothermic and increasing entropies of adsorption in the liquid phase can be found in a review by Anastopoulos and Kyzas[68]. Note that the less extreme $\Delta H_{ads}$ and $\Delta S_{ads}$ values for pyridine adsorption into ZSM-5 in the liquid phase compared to the vapor phase agree with results from our previous work, where molecularly adsorbed pyridine is easily removed from sites in the ATR-FTIR by flowing pure, room-temperature solvent through the cell, while in vacuum, molecularly adsorbed pyridine can be difficult to entirely remove below 200 °C[40].

In summary, a quantitative spectroscopic method is developed for measuring liquid-phase adsorption isotherms over porous materials, capable of distinguishing protonated from unprotonated, molecular adsorption sites. The length scale of the confinement effect is quantified in liquid water, where the pore-phase pyridine concentration sharply decreases in samples with increasing pore diameters from 7 to 15 Å, and the confinement effect is minimal in a 25 Å pore Si/MCM-41. These adsorption isotherms are used to estimate pore-phase standard-state chemical potentials and pore-phase activity coefficients from infinite dilution until pore-phase saturation. By comparing standard-state chemical potential differences between the zeolite and liquid phases to those obtained from vapor-liquid equilibrium, Haber cycles are constructed to illustrate quantitatively how the energetics of the external liquid-phase versus that of the pore-phase affect adsorption into porous materials in solvent. Adsorption isotherms on siliceous and aluminosilicate Beta zeolites are compared to estimate fundamental measures of apparent acidity in solvent-saturated Beta pores. DFT investigations reveal that pore-phase proton transfer occurs from a protonated water-cluster, a protonated acetonitrile dimer, and

from the Al-O-Si bridge-site in the case of 1,4-dioxane. Both experimentally-determined equilibrium constants for proton transfer and DFT simulations confirm increasingly favorable PTE in the order: water < acetonitrile < 1,4 – dioxane. Compared to the effect of the identity of the solvent, the zeolite framework structure causes minimal disparities in the proton transfer equilibrium constant. In addition to chemical potentials, activity coefficients, and apparent acidities, methods are outlined to estimate thermodynamic properties of the zeolite pore phase, including the change in solvation free energy ($\Delta G_{ads}$), enthalpy ($\Delta H_{ads}$), and entropy ($\Delta S_{ads}$) between the bulk solvent and the pore-phase. In water, $\Delta H_{ads}$ and $\Delta S_{ads}$ values for pyridine adsorption into Si/ZSM-5 are approximately −30 and −53 J mol$^{-1}$ K$^{-1}$, respectively. These estimates are compared to established vapor phase values to account for the lower magnitude of liquid-phase adsorption in zeolites.

## Methods

**Materials preparation**. NH$_4$/ZSM-5 (Si/Al = 11.5), NH$_4$/ZSM-5 (Si/Al = 40), NH$_4$/Beta (Si/Al = 12), H/Y (Si/Al = 15), H/Y (Si/Al = 2.6) were purchased from Zeolyst International. All zeolite samples were calcined in air at 500 °C for 10 h with a heating ramp of 1 °C min$^{-1}$ prior to use. Pyridine, acetonitrile, ethanol, and 1,4-dioxane were obtained from Thermo Fisher, and used without further purification. Siliceous Beta (Si/Beta) was synthesized by mixing 1 g of H/Beta (Si/Al = 12) with 25 mL of nitric acid, and heating for 24 h at 90 °C[71]. The resulting solid was washed and filtered via centrifugation for five cycles. Si/Beta was confirmed aluminum-free via X-Ray Fluorescence (XRF). Hydrophobic Si/Beta was synthesized in aqueous solution containing tetraethylorthosilicate (TEOS, Sigma-Aldrich, 98%) and tetraethylammonium hydroxide (TEAOH, Alfa Aesar, 35%). The mixture was placed in a 23 mL Teflon liner (A280AC, Parr). Hydrofluoric acid (HF, Acros, 48%) was manually added and mixed until a gel formed. The gel also contained siliceous zeolite Beta seeds that were synthesized via the method outlined in Camblor et al[72]. The resulting gel was transferred to a Teflon autoclave inside a stainless steel mantle which was heated in an oven at 413 K under static conditions for 20 days. The molar ratios of the gel components were 1:0.56:0.56:7.5 (SiO$_2$: TEAOH:HF:H$_2$O). Post synthesis Si/Beta was washed, filtered, and dried overnight at 353 K. The hydrophilic and hydrophobic nature of the two types of siliceous Si/ Beta is confirmed by transmission FTIR spectroscopy (Supplementary Fig. 37). A subsequent three-hour calcination at 853 K in air removed the SDA. Si-MCM-41 and SBA-15 samples with varying mesopore sizes were prepared by a method outlined by Grün et al. and Sayari et al. with minor modifications, respectively[73,74]. More detailed synthesis conditions are listed here for specific materials:

Si-SBA-15: Pluronic P-123 (Mn = 5800, Sigma Aldrich) was added to a mixture consisting of deionized (DI) water and HCl. The mixture was stirred at room temperature until a homogeneous solution was obtained. To this solution, TEOS was added, yielding a composition of 1 SiO$_2$:5.7 HCl:0.017 P-123:192.7 H$_2$O. The resultant solution was maintained at room temperature under static conditions for 20 h, followed by 24 h at 80 °C. Afterward, the solid was washed DI water, followed by drying at 80 °C overnight. The samples were calcined at 550 °C for 12 h with a ramping speed of 0.5 °C min$^{-1}$ under air flow to remove organic components from the solid.

Si-MCM-41 (2.5 nm pore): CTAB (n-hexadecyltrimethylammonium bromide, 99%, Sigma Aldrich) was dissolved in DI water. NH$_4$OH (29 wt %, Sigma Aldrich) and ethanol (EtOH) were added to the CTAB solution. To this mixture, TEOS was added, and the mixture was stirred for 2 h. The composition of the solution was 1 SiO$_2$:11 NH$_3$:0.3 CTAB:144 H$_2$O:58 EtOH. After stirring, the white precipitate was filtered and washed with methanol (MeOH)-H$_2$O solution (DI water: MeOH = 1:1, vol ratio), followed by drying at 80 °C overnight. Afterward, the sample was calcined at 550 °C for 12 h with a ramping speed of 0.5 °C min$^{-1}$ under air flow, in order to remove organic components from the solid.

Si-MCM-41 (1.5 nm pore): Synthesis procedure was the same as that of Si-MCM-41 with 2.5 nm mesopores, but decyltrimethylammonium bromide (98%, Sigma Aldrich) was used in place of CTAB and the solution had a composition of 1 SiO$_2$:11 NH$_3$:0.3 CTAB:144 H$_2$O:27 EtOH.

**Materials characterization**. N$_2$ physisorption with the t-plot method was employed to determine micropore volumes of zeolite samples (Supplementary Table 4) on a Micromeritics ASAP 2020 instrument. $^{29}$Si and $^{27}$Al magic angle spinning nuclear magnetic resonance (MAS-NMR) spectra of the zeolite samples were collected (Bruker DSX-200 spectrometer with a Bruker 7 mm MAS probe) by applying the following procedure: a 90° pulse of 4 microseconds followed by a strong $^1$H decoupling pulse. Sample spinning rate and the recycle delay were set at 4 kHz and 30 s, respectively. The $^{29}$Si and $^{27}$Al NMR spectra are included in Supplementary Fig. 19 and calculated Si/Al ratios estimated based on the $^{29}$Si NMR spectra were included in Supplementary Table 4. The Si/Zr ratio of Zr/Beta was determined to be 199 by ICP-AES. Powder X-ray Diffraction (XRD) patterns were

collected on a Bruker D8 Discover powder diffractometer with a Cu Kα source over the range 5°–50° with a step size of 0.025° and 2 s per step. XRD patterns of zeolite samples are included in Supplementary Fig. 20. Average mesopore volumes were estimated via the BJH method and isotherms for mesoporous materials are in Supplementary Figs. 21–23.

**Quantitative attenuated total reflection FTIR spectroscopy**. A homemade multiple reflection ATR flow cell was employed in the liquid-phase in-situ FTIR spectroscopic investigations with a 4 cm$^{-1}$ spectral resolution and 0.5 cm$^{-1}$ aperture[75]. A trapezoidal ZnSe ATR crystal with two 45° cuts was employed to provide six total internal reflections. A constant interferogram intensity of 5.0 (arbitrary units on an Agilent CARY 660 FTIR spectrometer) was achieved before depositing catalyst on the ZnSe crystal by adjusting the mirror between the ATR cell and the detector. The catalyst layer was directly deposited on the top side of the ZnSe ATR crystal by evaporating the solvent from a catalyst slurry. The seal between the ZnSe ATR crystal and the top plate is achieved by pressing an O-ring into a groove on the bottom surface of the top plate. The internal dimensions of the homemade ATR flow cell are 1.75 in. length × 0.5 in. width × 20 μm thickness for a total volume of $1.1 \times 10^{-2}$ cm$^3$. A piece of quartz microfiber filter paper (2.7 μm pore size, Whatman Inc.) was placed between the catalyst layer and the top plate to prevent catalyst powder from being washed out by the flow of the solvent. No loss of catalyst was observed in any experiments reported in this work for up to 24 h of continuous flow of solvents. In addition, no probe molecule adsorption was detected on the filter paper in control experiments devoid of catalyst. Solvents with or without dissolved probe molecules, e.g., pyridine, were introduced to the ATR cell via an HPLC pump (Waters 515) at a rate ranging from 0.2 to 1.0 mL min$^{-1}$.

The evanescent wave in the ATR-FTIR spectroscopy probes both the bulk liquid and the liquid-solid interface within its penetration depth from the top of the ZnSe crystal[76]. Thus, spectra contain signals from species both dissolved in the bulk solution and adsorbed on the catalyst. Spectroscopic features of the probe molecule, e.g., pyridine, in the bulk solution are removed by running a blank experiment devoid of a catalyst and performing a spectral subtraction. The subtraction method is further discussed in next subsection. Pyridine is known to be too bulky to enter zeolite pores with smaller than 10-membered ring openings, as demonstrated in our previous work[77]. A control experiment of pyridine adsorbed on a chabazite sample (H/SSZ-13) with 8-membered ring openings revealed the contribution of external acid sites to adsorbed pyridine bands was negligible, as confirmed by the lack of the band corresponding to BAS or pyridinium (1547 cm$^{-1}$) in our previous work[75].

**Methodology for obtaining liquid-phase adsorption isotherms**. Liquid-phase adsorption isotherms are conducted in the ATR-FTIR cell after depositing a catalyst film. A background spectrum is collected by flowing pure solvent until the spectrum equilibrates. The pure solvent feed is switched to a dilute pyridine in solvent solution, and spectra are collected until equilibrium is obtained. Increasingly concentrated pyridine solutions are fed through the ATR, waiting for equilibrium after every concentration increase (example set of spectra in Supplementary Fig. 6). In these spectra, the 1545 cm$^{-1}$ peak labelled B corresponds to BAS, peaks denoted M are MAS, and the 1490 cm$^{-1}$ peak is known to contain contributions from both protonated and molecular pyridine[19,78–80]. The 1545 and 1444 cm$^{-1}$ bands are used to quantify the Brønsted and molecularly adsorbed pyridine, respectively. Note that a fraction of the 1444 cm$^{-1}$ corresponds to bulk, liquid-phase pyridine (Supplementary Fig. 24, red dotted traces). Quantitative estimates of the amount of pyridine in zeolite pores are guided using batch, phase equilibria experiments in which 100 mg of catalyst is mixed overnight with 10 mL of a known concentration of pyridine (typically 0.05 M) in the solvent under study (Supplementary Fig. 25). The final state solution is filtered and the pyridine concentration is measured via gas chromatography (GC). The pyridine concentration in the zeolite pores is estimated via mass balance equations comparing the initial and final state bulk pyridine concentrations and the zeolite micropore volume measured via nitrogen adsorption. To account for varying catalyst film quality across different adsorption experiments in ATR-FTIR, the fraction of the IR evanescent wave measuring pyridine inside catalyst particles, referred to as $x_Z$, (opposed to pyridine in the external solvent, $1 - x_Z$) is estimated using Eq. 8, where $A_{cat}$ is the integrated area of the band at 1444 cm$^{-1}$ in the ATR-FTIR experiment containing catalyst (blue curve), $A_L$ is the integrated area of the band at 1444 cm$^{-1}$ in the ATR-FTIR experiment containing no catalyst (red dashed curve), $C_{Z, pore}$ is the concentration of pyridine in zeolite pores, and $C_L$ is the concentration of pyridine in the external solvent. A set of $C_{Z, pore}$ and $C_L$ values is obtained from the batch uptake experiment (Supplementary Fig. 25, final state). $A_L$ and $A_{cat}$ are obtained by measuring the integrated area of the band at 1444 cm$^{-1}$ in the ATR-FTIR devoid of catalyst and with catalyst, respectively, while using the final state pyridine concentration in the feed (Supplementary Fig. 25, where $C_L = C_{initial} - x$). Note that in the case of aluminum-form zeolites which contain both Brønsted and Lewis acid sites, the amount of pyridine on each type of site was deconvoluted using previously determined liquid-phase extinction coefficients[25]. Thus, $x_Z$ was calculated using Eq. 9 (rearranged version of Eq. 8). The $x_Z$ value was assumed to be constant over one experiment, and $C_Z$ was estimated using Eq. 10 (rearranged version of Eq. 8) for every data point on the isotherm. Note that performing a subtraction of the spectra in Supplementary Fig. 24 (blue minus red) requires one

to account for the volume that zeolite particles occupy within the ATR cell that is unoccupied during the control experiment devoid of catalyst (red). The fraction of the IR evanescent wave that samples zeolite particles is estimated by scaling $x_Z$ (Eq. 9) by one divided by the occupiable volume fraction of a zeolite framework (for pyridine) as provided by the Internal Zeolite Association (IZA, Eq. 11). This correction accounts for the fact that only a fraction of space occupied by zeolite crystals can accommodate adsorbates such as pyridine. Note that occupiable volumes are provided by IZA in terms of water accessibility. For FAU zeolites, the volume of sodalite cages (inaccessible to pyridine) are removed from the occupiable volume.

$$\frac{A_{cat}}{A_L} = \frac{x_Z C_{Z,pore} + (1 - x_Z)C_L}{C_L} \tag{8}$$

$$x_Z = \frac{C_L(A_{cat} - A_L)}{A_L(C_Z - C_L)} \tag{9}$$

$$C_Z = \left(\frac{C_L}{x_Z}\right)\left\{\frac{A_{cat}}{A_L} - (1 - x_Z)\right\} \tag{10}$$

$$\text{MAS subtracted peak area} = A_{cat} - A_L\left\{1 - \left(\frac{x_Z}{\text{percent occupiable volume}}\right)\right\} \tag{11}$$

**Liquid-phase adsorption thermodynamics**. The free energies of transfer for pyridine ($\Delta G_{ads}$) from bulk liquid solvent into zeolite pores are calculated using Eq. 12, where Z and L subscripts stand for the zeolite and bulk liquid phases, respectively. This equation is derived from equating the chemical potentials of the two phases in Eq. 13. Here, $\mu_i$ is the chemical potential, $\mu_i^*$ is the pseudochemical potential, and $C_i$ is the concentration of pyridine in phase $i$[81]. $\Delta G_{ads}$ is the free energy change associated with transferring a solute from a fixed position in the bulk liquid to a fixed position inside the zeolite pores, at constant temperature, pressure, and compositions, and as such includes all non-ideal molecular interactions in both phases. $\Delta G_{ads}$ can also be viewed as the difference in solvation free energy from an ideal gas of the solute to the bulk liquid and to the zeolite. In this context and for solution-phase adsorption, the solvation by zeolites is a combination of the zeolite pore walls and the solvent molecules in the pores.

$$\Delta G_{ads} = \mu_Z^* - \mu_L^* = RT \ln\left(\frac{C_L}{C_Z}\right) = \mu_{pyr,z}^\circ - \mu_{pyr,L}^\circ + RT \ln\left(\frac{\gamma_{Z,H}}{\gamma_{L,H}}\right) = \Delta H_{ads} - T\Delta S_{ads} \tag{12}$$

$$\mu_Z = \mu_L = \mu_Z^* + RT \ln C_Z = \mu_L^* + RT \ln C_L \tag{13}$$

$$\mu_i(T, P, C_i) = \mu_{i,H}^\circ(T, P, C_i = 1) + RT \ln\left(\gamma_{i,H}\frac{C_i}{C_i^\circ = 1}\right) \tag{14}$$

$$\mu_{pyr,L}^\circ + RT \ln\left(C_L \gamma_{L,H}\right) = \mu_{pyr,V}^\circ + RT \ln\left(\emptyset_i y_i P\right) \tag{15}$$

$$\mu_{pyr,L}^\circ + RT \ln\left(C_L \gamma_{L,H}\right) = \mu_{pyr,Z}^\circ + RT \ln\left(C_Z \gamma_{Z,H}\right) \tag{16}$$

In conventional solution thermodynamics treatment of phase equilibria and reaction kinetics[14], chemical potentials ($\mu_i$) are usually defined as a standard-state value ($\mu_{i,H}^\circ$) and a concentration-dependent term ($RT \ln C_i/C_i^\circ$), and deviations from such ideal behavior are then captured in the form of activity coefficients ($\gamma_{i,H}$); see Eq. 14. The connection between the two formalisms is made clear by the third equality in Eq. 12. The standard-state chemical potential is customarily extrapolated from the infinite dilution regime in liquid-phase adsorption isotherms, where Henry's law holds and $\gamma_{i,H} = 1$, to $C_i^\circ = 1$ M. Eqs. 15 and 16 define the vapor-liquid and zeolite-liquid-phase equilibria, respectively, used to calculate the differences in standard-state chemical potentials and pore-phase activity coefficients ($\gamma_{Z,H}$). The UNIQ-RK method in Aspen is used for generating P-x-y diagrams for calculating the difference in the liquid and vapor standard-state chemical potentials ($\mu_{pyr,L}^\circ$ - $\mu_{pyr,V}^\circ$) in Eq. 15. Liquid-phase activity coefficients obtained from ASPEN are based on mole fractions and Raoult's law (i.e., the standard state is taken to be pure pyridine liquid) and denoted by subscript R. They can be converted to the earlier definition based on concentrations and Henry's law. From P-x-y diagrams, liquid pyridine is found to be within the Henry's law regime at all concentrations used in the liquid-zeolite adsorption isotherms ($\gamma_{L,H} \approx 1$ in Eq. 16). The difference between the zeolite- and liquid-phase standard-state chemical potentials ($\mu_{pyr,Z}^\circ$ - $\mu_{pyr,L}^\circ$) in Eq. 16 equals $\Delta G_{ads}$ at infinite dilution and is estimated by using the point on the zeolite-liquid adsorption isotherms (the fitted Langmuir lines) nearest infinite dilution. Note that for Eqs. 12–16, the pyridine concentration in the pore phase ($C_Z$) is computed based on the total volume occupied by the zeolite framework. This is expressed mathematically in Eq. 17, where $\rho_Z$ is the overall density of the zeolite (including both framework atoms and empty pore space) estimated from the number of T atoms per 1000 Å$^3$ listed on the International Zeolite Association (IZA) website.

$$C_Z = C_{Z,pore} \times \frac{\text{micropore volume}}{(1/\rho_Z)} \tag{17}$$

**Computational details**. All DFT calculations for periodic systems were performed using VASP. The interaction between electrons and ions was modelled with the projected augmented wave (PAW) method with an energy cut-off of 400 eV[82,83]. Standard GGA PBE potentials were used for all elements and the DFT-D3 method was used to model adsorbate-zeolite dispersive interactions[84,85]. The first Brillouin zone was sampled at the Γ-point. A periodic three-dimensional, all-siliceous BEA structure, with lattice parameters 12.632 × 12.632 × 26.168 Å$^3$ was used in this work. The unit cell of H/Beta was then built by replacing one Si atom at the T9 lattice position with an Al atom. Solvent effects were considered by including several explicit solvent molecules in the pores of the zeolite unit cell. Equilibrium geometries were obtained from full structure optimization of low-energy configurations sampled from a 2-ps long AIMD trajectory at 373 K. For geometry optimizations, the force convergence tolerance was set to 0.02 eV Å$^{-1}$. We report only electronic energies, without thermal corrections, to prevent spurious low-frequencies from influencing the interpretation of our data (especially in the presence of solvent molecules). More details may be found in the Supplementary Discussion.

## Data availability
The source data underlying Figs. 2–4 and 7 and Supplementary Figs. 1–18, 24–31, 37 are provided as a Source Data file.

## Code availability
The Matlab code for fitting Langmuir Isotherms is provided as Source Data file.

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

## Acknowledgements

We acknowledge support from the Catalysis Center for Energy Innovation, an Energy Frontier Research Center funded by the U.S. Department of Energy, Office of Science, Office of Basic Energy Sciences under Award number DE-SC0001004.

## Author contributions

N.G. and B.X. designed experiments, N.G. conducted ATR experiments, H.C. synthesized porous catalysts, N.G., B.X. P.B., and H.L. analyzed experimental results. S.L., S.C., and D.V. designed and conducted DFT calculations, and analyzed the results. N.G., S.L., S.C., D.V., P.B., and B.X. co-wrote the paper.

## Competing interests

The authors declare no competing interests.
