## [Peer Review File · Nature Communications]

Reviewers' comments:

Reviewer #1 (Remarks to the Author):

This manuscript uses results from ATR-IR to calculate isotherms for pyridine adsorption in zeolites with various properties in water, acetonitrile, and 1,4-dioxane solvents. The isotherm results are used to estimate free energies of adsorption for molecular or Brønsted adsorption. DFT results indicate that Brønsted adsorption in water and acetonitrile is actually mediated by protonated solvent molecules, rather than Brønsted acid sites on the zeolite surface.

These mechanistic insights are novel, of interest to the community of liquid phase heterogeneous catalysis, and helpful for advancing thinking in the field of catalysis involving liquid molecules and Brønsted acids. The research presented in this manuscript is solid, and the authors have backed up their claims with AIMD simulations.

I recommend that this manuscript be published after the following issues are addressed:

-The novel finding in the manuscript is the identification of "specific acid sites," i.e., where the proton is solvated, and proton transfer to pyridine is solvent-mediated. This result seems buried, in part because solvent-mediated transfer is discussed with Brønsted adsorption, as if the two phenomena are the same. I recommend 1) to distinguish these two mechanisms, e.g., by adding onto Figure 1, and 2) to re-organize the manuscript text to emphasize this finding.

-The manuscript is difficult to read at times because of the need to frequently reference a rather dense Supporting Information document. It would greatly clarify the manuscript if the authors provided a summary in the main manuscript text about how the results from ATR-IR were translated into the various quantities that are reported.

-This manuscript could be made stronger by providing a more thorough citation of the literature. For example, the arguments about entropy (of the solvent having a dominating influence on the entropy of adsorption) and to some extent proton transfer coming from liquid molecules have been posed in the adsorption and catalysis literature by others.

-Page 9 "As the pore diameter decreases, the adsorption isotherms in Figure 2 become increasingly Langmuirian..." By visual inspection of Figure 2, all the isotherms look Langmuirian. Please clarify.

-The authors should clarify that pseudochemical potential means constant T, P, N.

-Figure 1 and similar figures: use different colored lines or different styles of lines to distinguish vapor versus solvent phase.

-Page 19: If water is not truly a statistical outlier, then a different word should be used.

-Page 19: Why is the comparison to ammonia relevant? Please provide some context.

-Page 24: "obtained estimated" seems to be a typo.

Reviewer #2 (Remarks to the Author):

This paper attempts to address a key challenge in catalysis which is understanding of solvent effects on adsorbates in porous media. The authors attempt is based on measuring adsorption isotherms and thermodynamic models. While this study is an interesting attempt at addressing this challenge, there are a number of issues that lead me to question the reliability of the presented results.

1) The authors argue that a Langmuir model can be used for determining equilibrium constants and all free energy arguments follow from being able to determine equilibrium constants. While I am not sure how large the error is, it is hard for me to understand why the assumptions from Langmuir are satisfied for adsorption in Zeolites. Ruthven (Nature Physical Science vol 232, page 71 (1971)) showed that even under ideal conditions adsorption in zeolites is not Langmuirian and that the corrections are important. Without showing that the approximations/errors introduced by the authors are not small, I fear publication is premature.

2) The authors should report error bars in the experimental data obtained by repeating the experiments multiple times. This is particularly relevant since some of the authors reported energies are quite small and possibly within the noise.

3) Why does it make sense to define a concentration in a zeolite? The concentration can only be an average value and the concentration is very different in different parts of the zeolite. The formalism appears too ad hoc and is likely not rigorous. I would also assume that the framework volume is solvent dependent.

4) The authors normalized the adsorption isotherms in figure 4. Since the adsorption isotherm should not be Langmuirian it should not be possible to normalize the isotherms.

Reviewer #3 (Remarks to the Author):

The paper reports a study on the acidity of porous catalysts in contact with a liquid solution.

Really, even if most of the data are probably good, the paper appears to me to be very complicated and unclear. Certainly, the topic is not simple, but the paper seems to me to be not well written and not useful in its present form.

Reviewer #4 (Remarks to the Author):

It is the aim of the authors to contribute to understand solvent effects on adsorption and protonation in porous catalysts. This is an important aspect in terms of processes proceeding on solid-liquid interfaces. The results should contribute to predict solvent behavior, and their impact on interactions between substrates and catalytic active sites. This work is a continuation of the former work of the authors (J. Catal. 2016, ACS Catal. 2018, Chem. Sci. 2018) using in ideal manner the pyridine-zeolite interaction for their studies. This is of course a well suited system in particular to study the interaction of pyridine with Brønsted acid sites and gain different physical parameters by analyzing respective isotherms. The question arises in which manner this approach is applicable to real reaction systems in which the reacting substrate is not pyridine. Considering the fact that the presented measurements are time-consuming, and required sophisticated equipment a general application is questionable. The authors should comment this in the conclusions.

As the authors demonstrated, their developed ATR-IR equipment is very well appropriate to study liquid-solid interactions. The experimental studies are qualified and the experimental results are analyzed properly.

Other comments:

In the abstract the authors should focus on the main aspects and results. In the present form, the first part is rather an introduction

Keywords: better ATR-FTIR or ATR-IR than FTIR only

Page 9: Langmuirian in shape – this is not an appropriate wording

Page 12 less enthalpically - this is also not an appropriate wording

Figure 4B: the units of K' values are missing

Page 19: DFT calculations of? . . . in solvent-saturated zeolite pores

Below we provide a point-by-point response to reviewers' comments, and the changes made are highlighted in the manuscript.

Reviewer 1

Comment 1: The novel finding in the manuscript is the identification of "specific acid sites," i.e., where the proton is solvated, and proton transfer to pyridine is solvent-mediated. This result seems buried, in part because solvent-mediated transfer is discussed with Brønsted adsorption, as if the two phenomena are the same. I recommend 1) to distinguish these two mechanisms, e.g., by adding onto Figure 1, and 2) to re-organize the manuscript text to emphasize this finding.

Response 1: We appreciate the reviewer's comment and agree that increased emphasis on the specific acid sites is warranted, particularly in the first half of the manuscript where the subject is not frequently discussed. A more explicit discussion of the proton transfer to the solvent clusters, and subsequently to pyridine will be included.

Action 1: Figure 1 is modified to increase to emphasis on the solvent-dependent specific acids and is included here as Figure R1:

Figure R1: Reaction coordinate diagram for a typical zeolite catalyzed, aqueous-phase reaction.

The discussion surrounding Figure 1 in the introduction on pages 4-5 is re-phrased to include the specific acid formation step added in the figure and emphasize its importance:

“ Gas phase reactions occurring in porous catalysts typically involve an initial adsorption into the pore phase (confinement), and a subsequent interaction with the acid site (either hydrogen-bonded or protonated), before the intrinsic activation step for the chemical reaction. The thermodynamics of liquid-phase reactions are more complex because both substrate and acid site energetics are affected by solvent interactions (Figure 1).^{1,2} The solvent identity affects the chemical potential of a proton through the formation of a solvent-dependent specific acid (A→B). In the case of a water-saturated zeolite pore, the proton spontaneously transfers to form a charged water cluster (Figure 1, brown). This water cluster is a specific acid with unique proton transfer properties compared to the framework-bound proton, which can have a significant impact on catalysis.³ The solvent identity also impacts the confinement step (B→C) through interactions that stabilize (or destabilize) the substrate (R) in the external liquid phase ($\Delta H_{\text{solv, B}}$) relative to the substrate in the pore phase ($\Delta H_{\text{solv, C}}$). Similarly, the hydrogen-bonding/protonation step (C→D) can be affected by the solvent identity because the structure of the specific acid is disrupted upon proton transfer. In this step, a solvent differentially affects the stability of the hydrogen-bonded or protonated adduct ($\Delta H_{\text{solv, D}}$) relative to the solvated proton and the pore-phase substrate ($\Delta H_{\text{solv, C}}$). The propensity for the proton to transfer to the substrate is affected by solvent interactions with all three of these species and is a major topic of investigation in this work. The solvent may also affect the intrinsic reaction barrier ($E_{\text{a,int}}$, step D→TS) by stabilizing the transition state ($\Delta H_{\text{solv, TS}}$) relative to the reactant adduct in step D.”

Increased emphasis about specific acids are included in multiple locations in the manuscript:

In the abstract:

“Density functional theory and ab initio molecular dynamics simulations are employed to investigate the preferred structure of a proton in the vicinity of co-adsorbed, pore-phase solvent molecules, including the formation of a protonated water-cluster, a protonated acetonitrile dimer, and zeolite framework-bound protons in the case of non-polar solvents. Both computational and experimental methods result in increasingly favorable pore-phase proton transfer to pyridine in the order: water < acetonitrile < 1,4 – dioxane.”

In the final paragraph of the introduction:

“For the solvents studied in this work, the free energy of pyridine protonation becomes more negative with decreasing solvent polarity, or dielectric constant. Computational investigations of the preferred structures of solvent-dependent specific acids are used to estimate the pore-phase proton transfer energy (PTE) to pyridine using density functional theory (DFT). Both DFT calculations and experimental adsorption isotherms confirm increasingly favorable PTE in the order: water < acetonitrile < 1,4 – dioxane.”

After defining K_{conf} and K_{prot} in the results section on pages 16-17:

“The confinement equilibrium constant (K_{conf}) involves liquid-phase pyridine entering the zeolite pore and displacing pore-phase solvent into the bulk liquid-phase (Equation 4), while the protonation equilibrium constant (K_{prot}) reflects proton transfer from the solvent-dependent specific acid to *pore-phase* pyridine (Equation 5). Further discussion about the stability of protons in these specific acid solvent-clusters will be discussed alongside density functional theory (DFT) calculations later in this work.”

In the conclusion section:

“Adsorption isotherms on siliceous and aluminosilicate Beta zeolites are compared to estimate fundamental measurements of apparent acidity in solvent-saturated Beta pores. DFT investigations reveal that pore-phase proton transfer occurs from a protonated water-cluster, a protonated acetonitrile dimer, and from the Al-O-Si bridge-site in the case of 1,4-dioxane. Both experimentally-determined equilibrium constants for proton transfer and DFT simulations confirm increasingly favorable PTE in the order: water < acetonitrile < 1,4 – dioxane.”

Comment 2: The manuscript is difficult to read at times because of the need to frequently reference a rather dense Supporting Information document. It would greatly clarify the manuscript if the authors provided a summary in the main manuscript text about how the results from ATR-IR were translated into the various quantities that are reported.

Response 2: We appreciate the reviewer’s comment and agree that more detail about the quantification method would be a worthwhile addition to the main text. Since the complete methodology described in the SI would require a significantly longer main text (by about 20-25%), we include an abbreviated version of the methods.

Action 2: An abbreviated version of the experimental methods relevant to the main manuscript is included as its own section prior to Results and Discussion:

2. Abbreviated Experimental Methods

Liquid-phase in-situ ATR-FTIR spectra are collected in a homemade multiple reflection ATR flow-cell loaded with a catalyst layer atop a ZnSe crystal.⁴ A background spectrum is collected by flowing pure solvent until the spectrum equilibrates. The pure solvent feed is switched to a dilute pyridine in solvent solution, and spectra are collected until equilibrium is obtained. Increasingly concentrated pyridine solutions are fed through the ATR, waiting for equilibrium after every concentration increase. ATR-FTIR peak areas are converted into adsorbed concentrations via previously determined liquid phase pyridine extinction coefficients,⁵ as well as a simple phase equilibrium experiment involving mass balances and gas chromatography (GC) that is fully discussed in Supplementary Information (SI 1.5). Several experimental considerations are accounted for, including distinguishing the spectroscopic features of pyridine in the bulk

solution from those of adsorbed pyridine, confirming that adsorption on the external zeolite surface is negligible compared to intraporous adsorption, and accounting for varying catalyst film quality across experiments. Experimental adsorption isotherm data points are fitted to a Langmuir model with a few exceptions mentioned in the results section. However, thermodynamic properties in this work are typically estimated from adsorption isotherms near infinite dilution (in the Henry's Law regime), where either a fitted adsorption model or experimental data points near infinite dilution may be used. Thus, thermodynamic estimates are not dependent on the applicability of any specific adsorption model. More detailed experimental and data analysis procedures are included in SI 1.3-1.5.

Comment 3: This manuscript could be made stronger by providing a more thorough citation of the literature. For example, the arguments about entropy (of the solvent having a dominating influence on the entropy of adsorption) and to some extent proton transfer coming from liquid molecules have been posed in the adsorption and catalysis literature by others.

Response 3: We appreciate the reviewer's comment and agree that more discussion from the literature on these topics should be included.

Action 3: Further discussion about entropy of adsorption is included on pages 25-26 after the entropy of pyridine adsorption into Si/ZSM-5 is estimated:

“Note that the weakened adsorption entropy loss in the liquid-phase has been observed in the literature, even to the extent where the entropy increases upon confinement.⁶ In the vapor phase, adsorption is most commonly an exothermic process and the decrease in translational/rotational degrees of freedom typically leads to a decrease in entropy. In liquid phase adsorption, an increase in entropy and even endothermicity is more common than in the vapor phase. Examples are prevalent in liquid water, such as the adsorption of ions from aqueous solution where the energy required to disrupt the hydration sphere exceeds the stabilization due to bonding to the solid surface.⁷⁻⁸ Increases in adsorption entropy from the aqueous phase is common in the adsorption of large-molecular dyes onto solid surfaces, as a large number of solvent molecules are displaced per mole of adsorbate.^{7,9} Positive entropies can also result in the case of adsorption of hydrophilic substrates (e.g., glucose) from aqueous solution. Hydrogen-bonding interactions of hydrophilic functional groups with liquid water can be disrupted upon confinement into microporous materials such as zeolites. This was observed in the case of glucose adsorption from aqueous solution into zeolite Beta.⁶ Further examples of endothermic and increasing entropies of adsorption in the liquid phase can be found in a review by Anastopoulos and Kyzas.⁷ Note that the less extreme ΔH_{ads} and ΔS_{ads} values for pyridine adsorption into ZSM-5 from the liquid phase compared to the vapor phase agrees with results from our previous work, where pyridine is easily removed from molecular adsorption sites in the ATR-FTIR by flowing pure, room-temperature solvent through the cell, while in vacuum molecularly adsorbed pyridine can be difficult to entirely remove below 200 °C.¹⁰

Further discussion about proton transfer is included after discussing the impact of confinement (K_{conf}) on protonation measurements on pages 20-21:

“In homogeneous liquid phase catalysis by BAS, the effect of the solvent on the relative stability of protons versus protonated transition states has been demonstrated to affect reaction rates and selectivities.³ This shift in the apparent rate constants is prevalent in work by the Dumesic group, where BAS catalyzed rates of 1,2-propanediol to propanal, and cellobiose to glucose increased in gamma valero lactone (GVL) compared to water due to the relative stabilization of the proton compared to the protonated transition state.¹¹ This idea is discussed for a general reaction in Figure 1, where the apparent and intrinsic activation barriers are affected by solvent interactions. More recent work of BAS catalyzed dehydration and hydrolysis reactions studied how the energetics of proton transfer to reactants with varying hydroxyl groups densities (hydrophilicity) were affected by the composition of organic solvents mixed with water. Organic co-solvents caused water-enriched domains to form in the vicinity of reactant hydroxyl groups, increasing hydroxyl group hydrogen-bonding strength with neighboring water molecules, and leading to stabilized proton transfer and protonated transition states compared to pure water.¹² An exhaustive study of mixtures of water with either DMSO, dioxane, THF, GVL, or acetonitrile demonstrated that molecular-scale interactions form multicomponent solvent clusters with highly varying proton affinities compared to either pure component. The heat of mixing of the two solvent components was proposed as a metric to describe the relative proton versus transition state stabilities, and thus the reactivities of a series of reactants of varying hydrophilic character.³ Concerning experimental methods, in recent work from the Shanks group, high-resolution MAS NMR was used to study mixed solvent interactions with BAS at the solid-liquid interface over supported sulfonic acid materials as well as H/ZSM-5.¹³ The technique was able to distinguish bulk water from water interacting with the BAS, and used water’s chemical shift as a metric of the relative acidity over the composition range in a mixture with d_6 -DMSO.”

Comment 4: -Page 9 "As the pore diameter decreases, the adsorption isotherms in Figure 2 become increasingly Langmuirian..." By visual inspection of Figure 2, all the isotherms look Langmuirian. Please clarify.

-The authors should clarify that pseudochemical potential means constant T, P, N.

-Figure 1 and similar figures: use different colored lines or different styles of lines to distinguish vapor versus solvent phase.

-Page 19: If water is not truly a statistical outlier, then a different word should be used.

-Page 19: Why is the comparison to ammonia relevant? Please provide some context.

-Page 24: "obtained estimated" seems to be a typo.

Response 4: We appreciate the reviewer’s comments. In Figure 2 (reproduced below), the concentration of pore-phase pyridine increases as the adsorbent pore-size decreases. This effect is primarily attributed to increasing van der Waals stabilization with the pore walls (increasing contact area). We believe that the stronger pyridine interactions with smaller pores leads to better agreement with the assumptions of the Langmuir model than for the larger, mesoporous materials.

The liquid phase isotherms do not satisfy Langmuir assumptions completely, but some of the Langmuir assumptions, namely the identical adsorption site and the lack of adsorbate-adsorbate interactions assumptions, are better in the small-pore cases. For further discussion about the applicability of the Langmuir model, see Response 1 to Reviewer 2's comments.

We provide a visual interpretation in Figure 2C to explain why the small pore samples improve these Langmuir assumptions. With the increasing pore diameter, a second or more "layers" can form, leading to more than one dominant type of interactions that pore-phase pyridine molecules can experience. With increasing pore diameter, pyridine experiences more effective interactions with neighboring pyridine or solvent molecules and less direct contact with pore walls (the equivalent of the surface in the Langmuir model). In the case of the small pore adsorbents, where the pore diameter is comparable to the size of pyridine molecules, the pyridine-pore wall interactions are strong, and the pyridine-solvent and pyridine-pyridine contact area is reduced relative to the mesoporous materials. It is this increase in the adsorbate-surface interaction strength relative to the adsorbate-adsorbate and the adsorbate-solvent interaction strength that improves the applicability of the Langmuir assumptions.

Figure 2: A): Pyridine adsorption isotherms on adsorbents of varying average pore diameters in liquid water in ATR-FTIR at 20 °C. The pore-phase pyridine concentration on MAS is denoted by “+” signs and the pyridinium is denoted by dots. The sum of the pyridine on BAS and MAS is denoted with a dashed line for H/USY. B): The pore-phase pyridine concentration with 0.1 M pyridine in the external solvent as a function of pore diameter. The ATR-FTIR spectra for the data in Figure 2 can be found in Supporting Information Figures S8-S13. Note that aluminosilicate FAU is used because siliceous FAU cannot be synthesized. C): Depiction of pyridine adsorption in micropores and mesopores where molecules of identical color experience similar adsorption energetics.

The comment about ammonia in this work is reproduced here for convenience:

“This is not surprising in the case of water, as the proton affinity of a water trimer is already equal to that of **ammonia**,¹⁴ *ca.* 9 eV, and the facile formation of hydrated H_3O^+ in zeolites has been extensively discussed in the literature.¹⁴⁻¹⁸ A single acetonitrile molecule does not become spontaneously protonated by the Al-O-Si bridge BAS, while its dimer is basic enough to spontaneously abstract the proton to form $[(\text{CH}_3\text{CN})_2\text{H}]^+$.”

This comment was primarily used to draw a quantitative comparison between the proton stabilization brought about by the water clusters (the specific acid in pore-phase water) to a well-known base. As discussed in the context of Table 2 (and the reviewer’s “outlier” comment), the proton transfer equilibrium constant is lower for water than for all other solvents, indicating the strong stability of the proton in the delocalized water cluster. The protonation of ammonia is a

well-known phenomenon that provides the reader with a rough understanding of the *magnitude* of the proton-stabilization that the water cluster is imparting. We felt this was a good reference for comparison because while we provide a quantitative sense of the differences in proton transfer across solvents, it is difficult for the reader to predict the magnitude of stabilization of a specific pore-phase solvent's behavior, particularly within the micropore environment, where it is unclear how or whether common bulk properties of solvents apply (e.g., density, dielectric constant, viscosity, etc.).

Table 2. Confinement and Proton Transfer Equilibrium Constants Near Infinite Dilution in Solvent

Solvent (Zeolite)	$K_{\text{conf}} K_{\text{prot}}$	K_{conf}	K_{prot}	ϵ
1,4 – Dioxane (H/Beta)	37500	9.1	4120	2
Ethanol (H/Beta)	21800	31	703	25
Acetonitrile (H/Beta)	13400	158	85	38
Water (H/Beta)	5610	1980	2.8	80
Water (H/Y)	3580	575	6.3	80

Action 4: Further discussion is included on the improved Langmuir assumptions in the case of small diameter adsorbents on pages 10-11:

“As the pore diameter increases in the 2.5 to 6.5 nm range, pyridine molecules only lose marginal contact area with pore walls, which start to resemble a flat surface. As the pore diameter decreases to that of zeolites ZSM-5 and Beta, the adsorption isotherms in Figure 2 increasingly resemble Langmuir isotherms, as pore-phase pyridine molecules experience a relatively strong interaction with pore walls. As the diameter increases to the extent that a second or more “layers” are sterically possible (Figure 2C), pore-phase pyridine molecules can experience more than one type of dominant interactions and interact predominantly with neighboring pyridine or solvent molecules at the expense of direct contact with pore walls (the equivalent of the surface in the Langmuir model). The increased adsorbate-surface interaction strength for Beta and ZSM-5 relative to the adsorbate-adsorbate and the adsorbate-solvent interactions for mesoporous materials results in isotherms with greater Langmuir character. Both the identical adsorption site and the non-interacting adsorbate assumptions are more valid for pore diameters of similar dimensions to the adsorbate. Further discussion and corroboration of the applicability of the Langmuir model to ZSM-5 and Beta isotherms in water can be found in the SI surrounding Equations S12-S13 and Table S3. Note that several thermodynamic properties can be estimated from the isotherms in Figure 2A. The free energy of transfer (ΔG_{ads}) for the adsorption isotherms in Figure 2A are

calculated using the relative concentrations of pyridine in the external liquid (“L”) and the pore phase (“Z”) using Equation 1 at constant temperature and pressure (T and P , SI Figure S14, and SI 1.5). Note that Equation 1 and all thermodynamic properties computed in this work are not

$$(1) \quad \Delta G_{\text{ads}} = -RT \ln \left(\frac{C_Z}{C_L} \right)$$

dependent on the use of the Langmuir model. For example, the adsorption free energy can be calculated for each data point along the adsorption isotherm via Equation 1. Additional fundamental thermodynamic properties of the pore phase computed from Figure 2A include the pore-phase pyridine standard-state chemical potentials (μ_Z°) and pore-phase activity coefficients (γ_Z), which are listed in SI Table S2 and Figure S15, respectively. Note that all pore-phase concentrations and thermodynamic properties computed in this work are ensemble averages, and zeolite pores do consist of multiple adsorption sites. However, this situation is not unlike that of a micro-heterogeneous solution, where aggregation and molecular clustering leads to variations in local concentration and modifies the environment surrounding any particular solute species. This is also the quantity conventionally used in the analysis of chemical kinetics, even for catalysts exhibiting high microheterogeneity, i.e., a total surface concentration of CO on supported metal catalysts with a diversity of adsorption sites. A full analysis and discussion of these thermodynamic properties can be found in the SI, alongside additional pyridine adsorption isotherms for several H/ZSM-5 and H/Beta samples with varying Si/Al ratios and hydrophilic/hydrophobic textures (Figure S16).”

All chemical potentials and pseudochemical potentials in this work are for constant T and P , which is noted on page 11:

“The free energy of transfer (ΔG_{ads}) for the adsorption isotherms in Figure 2A are calculated using the relative concentrations of pyridine in the external liquid (“L”) and the pore phase (“Z”) using Equation 1 following a pseudochemical potential formalism at constant temperature and pressure (T and P , SI Figure S14, and SI 1.5).”

$$(1) \quad \Delta G_{\text{ads}} = -RT \ln \left(\frac{C_Z}{C_L} \right)$$

Clarification of the comment regarding ammonia is included on page 21-22:

“The spontaneous protonation of water to form the protonated water cluster specific acid is highly favorable. As a point of reference, the proton affinity of a water trimer is similar to that of ammonia,¹⁴ *ca.* 9 eV, and the facile formation of hydrated H_3O^+ in zeolites has been extensively discussed in the literature.¹⁴⁻¹⁸

The word “outlier” was removed from page 19. The typo on page 24 is fixed. We thank the reviewer for his observation.

Reviewer 2

Comment 1: 1) The authors argue that a Langmuir model can be used for determining equilibrium constants and all free energy arguments follow from being able to determine equilibrium constants. While I am not sure how large the error is, it is hard for me to understand why the assumptions from Langmuir are satisfied for adsorption in Zeolites. Ruthven (Nature Physical Science vol 232, page 71 (1971)) showed that even under ideal conditions adsorption in zeolites is not Langmuirian and that the corrections are important. Without showing that the approximations/errors introduced by the authors are not small, I fear publication is premature.

Response 1: We appreciate the reviewer’s comment and understand the skepticism about the applicability of the Langmuir isotherm model used in this work. We agree that liquid-phase adsorption of pyridine into zeolites is not a situation traditionally characteristic of a Langmuir isotherm because the liquid-phase is almost always non-ideal and involves significant neighbor-neighbor interactions (both in the bulk and within the pore). Also, similar to the reviewer’s Comment 3, the adsorption sites are not identical, and the reported thermodynamic properties are ensemble average properties over all sites (we will address this in Response 3).

We stress that the use of a Langmuir model is not necessary to estimate the thermodynamic properties (ΔG_{ads} , ΔH_{ads} , ΔS_{ads} , γ_z , $\mu_{\text{pyr,Z}}^{\circ}$) in this work. Equilibrium constants can be computed at each measured data point based on the ratio of the pore-phase and the liquid-phase concentrations

Figure 3A: Pyridine adsorption isotherms on Si/Beta (hydrophilic) at 20 °C in various solvents.

(Equation 1). In other words, ΔG_{ads} can be calculated discretely for every individual data point using the two concentrations in Equation 1. Note that the ratio of concentrations should be constant (within experimental uncertainty) near the origin (i.e., within the Henry’s law regime). As all the thermodynamic properties estimated in this work can be calculated using this ratio of concentrations, the Langmuir model is merely one model that can be used to fit the data to estimate the equilibrium constant in the Henry’s law regime. Alternatively, we could have simply averaged the individual equilibrium constants at the lowest few

$$(1) \quad \Delta G_{\text{ads}} = -RT \ln \left(\frac{C_Z}{C_L} \right)$$

concentrations (linear fit in the Henry's law regime), or used any other isotherm model that has the correct asymptotic behavior at infinite dilution (i.e., reduces to Henry's law). Our free energy arguments therefore do not depend on the validity of the Langmuir model. We agree with the reviewer that this needs to be made clearer, as the reviewer is correct that the Langmuir model is not fully applicable to liquid-phase pyridine adsorption.

The reviewer's comment about the approximations/errors will be addressed in Comment 2, below.

Action 1: Comments about how the Langmuir model was used in this work, and its applicability (or lack thereof) to the adsorption isotherms in this work are included on page 8:

“However, thermodynamic properties in this work are typically estimated from adsorption isotherms near infinite dilution (in the Henry's Law regime), where either a fitted adsorption model or experimental data points near infinite dilution may be used. Thus, thermodynamic estimates are not dependent on the applicability of any specific adsorption model. More detailed experimental and data analysis procedures are included in SI 1.3-1.5.

Also on page 11 after discussing the isotherms in Figure 2:

“The free energy of transfer (ΔG_{ads}) for the adsorption isotherms in Figure 2A are calculated using the relative concentrations of pyridine in the external liquid (“L”) and the pore phase (“Z”) using Equation 1 at constant temperature and pressure (T and P , SI Figure S14, and SI 1.5). Note that

$$(1) \quad \Delta G_{\text{ads}} = -RT \ln \left(\frac{C_Z}{C_L} \right)$$

Equation 1 and all thermodynamic properties estimated in this work are not dependent on the use of the Langmuir model. For example, the adsorption free energy can be estimated discretely for every individual point using the relative concentrations via Equation 1, or via any applicable fitted adsorption model. Additional fundamental thermodynamic properties of the pore phase estimated from Figure 2A include the pore-phase pyridine standard-state chemical potentials (μ_Z°) and pore-phase activity coefficients (γ_Z), which are listed in SI Table S2 and Figure S15, respectively.”

More comments on the applicability of the Langmuir model are included in Response 4 to Reviewer 1.

Comment 2: The authors should report error bars in the experimental data obtained by repeating the experiments multiple times. This is particularly relevant since some of the authors reported energies are quite small and possibly within the noise.

Response 2: We appreciate the reviewer's comment and agree that an estimate of the error involved in the adsorption isotherms is necessary. In the Supporting Information, the reproducibility of pyridine adsorption on Si/Beta in water is analyzed with five identical trials

(Figure S21 reproduced below). For this case, the average and standard error of the mean of K' values are 207 ± 9 , with a standard deviation that is roughly 10 % of the mean value.

Figure S21: Liquid phase adsorption isotherms of pyridine into Si/Beta (hydrophilic) at 20 °C repeated 5 times. The control experiment of the empty cell area in black (bulk pyridine contribution).

exceeds the roughly 5.5 Å pore diameter of ZSM-5, and would likely incur an energetic penalty for the required zeolite deformation. As the molecular diameter of the alcohol adsorbates in Figure R1 increase from cyclopentanol, to cyclohexanol and cycloheptanol, the pore-phase adsorbate concentration decreases, likely due to the increasing zeolite energetic deformation penalty. This simple adsorption experiment demonstrates that the liquid-phase adsorption technique used in this work has sufficient sensitivity to capture this intuitive effect.

Error analysis for quantitative pyridine adsorption on zeolites was also conducted in a previous work,⁵ where integrated molar extinction coefficients (IMECs) were determined for pore-phase pyridine on both Brønsted acid sites and molecular adsorption sites in three common solvents: water, ethanol, and acetonitrile. The error associated with converting the ATR-FTIR peak area into the adsorbed quantity is reported in Table R1 below.⁵ Note that standard deviations on the IMECs range between 8 and 18 %.

In recent unpublished work, the ATR-FTIR was shown to sensitively measure the quantity of adsorbed cyclo-alcohols with varying size in ZSM-5 and Beta zeolites in liquid water (Figure R1, unpublished results for review only). For reference, cyclohexane, which has a molecular diameter of roughly 6 Å,

Table R1. Summary of Adsorbed Pyridine Extinction Coefficients in three solvents

Solvent	Brønsted IMEC ^a	Lewis IMEC ^a
Acetonitrile	1.98 ± 0.16	2.53 ± 0.38
Ethanol	2.98 ± 0.49	$\sim 2.2^b$
Water	2.55 ± 0.28	2.27 ± 0.41

^aAll values in $\text{mmol}/g_{\text{cat}}/\text{area}$.

^bAbsolute intensities of Lewis bands in ethanol too weak to obtain a reliable estimate of error bars.

Figure R1: Liquid-phase cyclopentanol, cyclohexanol, and cycloheptanol adsorption isotherms over (A) Si/ZSM-5 and (B) Si/Beta in liquid water.

Action 2: A comment about the repeated Si/Beta trials in water and previous statistics of pyridine extinction coefficients in solvent is included in the main text:

“The adsorption isotherm on Si/Beta is most dramatic in the case of water, which has a Langmuir adsorption constant (K') equal to 207 ± 9 (Figure 3A legend). This experiment is repeated multiple times in Figure S21, resulting in a standard error of the mean equal to 9. In a previous work, pyridine and pyridinium extinction coefficients on zeolites in solvent were shown to have standard deviations ranging from 8 to 18 % depending on the solvent of interest.⁵”

Comment 3: Why does it make sense to define a concentration in a zeolite? The concentration can only be an average value and the concentration is very different in different parts of the zeolite. The formalism appears too ad hoc and is likely not rigorous. I would also assume that the framework volume is solvent dependent.

Response 3: We agree with the reviewer that the interior pore space of a zeolite is non-uniform and that local concentrations likely vary. However, this situation is not unlike that in a microheterogeneous solution, where aggregation and molecular clustering leads to variations in local concentration and modifies the environment surrounding any particular solute species.³ There are two reasons why in both cases it may still be desirable to define an overall concentration. First, while techniques exist for probing sorbate siting and local structuring, it is often only feasible to reliably obtain an overall loading/concentration. This is also the quantity conventionally used in quantitative analysis of chemical kinetics,¹¹ even for catalysts exhibiting high microheterogeneity, i.e., a total surface concentration of CO on supported metal catalysts with a diversity of sites. These non-ideal effects, due to strong, specific sorbate-sorbent, sorbate-sorbate interactions, are then captured in the form of activity coefficients. We do not believe that our treatment is ad-hoc or not rigorous, as treating the zeolite phase using solution thermodynamics is the basis for successful

theories such as the Real / Ideal Adsorbed Solution Theory (RAST/IAST), and the pseudochemical potential formalism is well known and can be found in standard statistical thermodynamics textbooks.¹⁹ Second, even in the case where siting probability can be obtained [e.g., from molecular simulations,^{20,21}], it is still possible and productive to cast the rate equations to the more familiar form in terms of overall concentrations.²¹ We also note that the heterogeneity of adsorption sites in zeolites also implies that the gas phase adsorption isotherms can only lead to average values of thermodynamic variables, which does not prevent these measurements from obtaining rich information regarding the solid-gas interactions.

We appreciate the reviewer's final suggestion that the zeolite framework volume is solvent dependent. We would like to clarify that we do not intend to suggest that the ATR-FTIR method developed in this work is a reliable method to determine the micropore volume of micro/mesoporous materials. In a sense, the amount of pyridine trapped in the micropores in a given solvent is directly measured by our adsorption isotherm experiments in Figure 3A of the manuscript (reproduced below for convenience), where the identity of the solvent is shown to have a strong impact on the pyridine adsorption isotherms in a common zeolite (Si/Beta).

Figure 3A: Pyridine adsorption isotherms on Si/Beta (hydrophilic) at 20 °C in various solvents.

As a function of the pyridine concentration, as was done in the isotherms in this work. By this logic, we view the reviewer's final comment as another way of interpreting the isotherms in Figure 3A.

Action 3: Discussion about the ensemble average pore-phase concentration is included in the first results and discussion paragraph after explaining Figure 2:

“Note that all pore-phase concentrations and thermodynamic properties estimated in this work are ensemble averages, and zeolite pores do consist of multiple adsorption sites. However, this situation is not unlike that of a micro-heterogeneous solution, where aggregation and molecular

impact on the pyridine adsorption isotherms in a common zeolite (Si/Beta). However, the micropore volume occupied by pyridine is a result of the final equilibrium state determined by equivalent chemical potentials in the liquid and the pore-phase, rather than a steric argument where solvent molecules are filling sections of the pore that pyridine could no longer occupy. It is also difficult to define what particular condition (the pyridine concentration in solvent) should be used to measure the pore volume in solvent. At the dilute pyridine limit, we are not measuring the maximum pore-volume that the pyridine can occupy, while at the concentrated limit, the pure, solvent-free state is not accounting for interactions with solvent. The only remaining option is to measure the occupied volume by pyridine as a

clustering leads to variations in local concentration and modifies the environment surrounding any particular solute species.³ This is also the quantity conventionally used in the analysis of chemical kinetics,²² even for catalysts exhibiting high microheterogeneity, i.e., a total surface concentration of CO on supported metal catalysts with a diversity of adsorption sites.”

Comment 4: The authors normalized the adsorption isotherms in figure 4. Since the adsorption isotherm should not be Langmuirian it should not be possible to normalize the isotherms.

Response 4: We appreciate the reviewer’s comment. Related to the comments in Response 1, the thermodynamic properties estimated in this work do not depend on the applicability of the Langmuir model. The isotherms are normalized with respect to the saturation loading, the existence of which can be confirmed for the concentrations at which our experiments were performed.

Action 4: The statement concerning normalization on page 16 is clarified by mentioning the saturation loading:

“The isotherms are normalized according to the saturation loading on BAS to highlight the slight change in the curve shape of the isotherms.”

Reviewer 3

Comment 1: The paper reports a study on the acidity of porous catalysts in contact with a liquid solution. Really, even if most of the data are probably good, the paper appears to me to be very complicate and unclear. Certainly, the topic is not simple, but the paper seems to me to be not well written and not useful in its present form.

Response 1: We appreciate the reviewer's comment. Although we tried to make the paper as accessible as possible, the subject matter of this work is quite fundamental and requires significant detailed thermodynamics discussions. We believe we struck a balance between technical rigor and accessible reading, but if the reviewer has more specific criticisms, we would be happy to make necessary changes.

Action 1: The main text is edited to improve its readability in several ways. Major edits include:

- 1) An abbreviated experimental and analysis section that explains how isotherms are performed using the liquid phase FTIR technique, and how isotherm data is used to estimate thermodynamic properties.
- 2) A modification of Figure 1 to include proton transfer from the framework Al-O-Si site to the specific acid solvent-cluster. Introducing the specific acid idea earlier improves the clarity of the pore-phase proton transfer discussion later in the work.
- 3) Explicit discussion about the assumptions of the Langmuir adsorption model and why estimates of thermodynamic properties in this work are not dependent on the applicability of the Langmuir model to liquid phase adsorption isotherms.
- 4) Additional discussion from the literature on liquid phase proton transfer and solvent effects on the entropy of adsorption.
- 5) A modified abstract with an increased focus on results.

Reviewer 4

Comment 1: It is the aim of the authors to contribute to understand solvent effects on adsorption and protonation in porous catalysts. This is an important aspect in terms of processes proceeding on solid-liquid interfaces. The results should contribute to predict solvent behavior, and their impact on interactions between substrates and catalytic active sites. This work is a continuation of the former work of the authors (J. Catal. 2016, ACS Catal. 2018, Chem. Sci. 2018) using in ideal manner the pyridine-zeolite interaction for their studies. This is of course a well suited system in particular to study the interaction of pyridine with Brønsted acid sites and gain different physical parameters by analyzing respective isotherms. The question arises in which manner this approach is applicable to real reaction systems in which the reacting substrate is not pyridine. Considering the fact that the presented measurements are time-consuming, and required sophisticated equipment a general application is questionable. The authors should comment this in the conclusions. As the authors demonstrated, their developed ATR-IR equipment is very well appropriate to study liquid-solid interactions. The experimental studies are qualified and the experimental results are analyzed properly.

Response 1: We appreciate the reviewer's comment and agree that commenting on the applicability of the technique to reacting systems other than pyridine is a meaningful addition. Most of the limitations of the technique are those of ATR-FTIR itself, rather than the adsorption/thermodynamic methods outlined in this work. The most stringent limitation of ATR-FTIR is keeping the reaction/adsorption temperature below the solvent boiling point. Note that a diamond reflective element may be used in conjunction with back-pressure to raise the temperature above the boiling point. The same methodology outlined in this work and in the previous works the reviewer cited can be applied to any reflective element material.

The other limitations of ATR-FTIR under reaction conditions are the same as for nearly any spectroscopic technique: 1) that you can spectroscopically distinguish the species of interest, e.g., pyridine vs. pyridinium in this case, or reactant vs. intermediate, and 2) that the species of interest (often intermediates) are stable enough to exist in an observable quantity. Note that some improvement in the surface-sensitivity of ATR-FTIR has been made by implementing modular excitation spectroscopy (MES) with phase-sensitive detection (PSD).²³⁻²⁴ Further, the liquid-phase equilibrium-adsorption method outlined in this work could be used with other forms of spectroscopy in cases when infrared is unable to distinguish intermediates (the necessary modified analytical equipment would be required).

The criticism that the technique is time-consuming is a valid point. Note however, the adsorption isotherm method in this work is designed for equilibrium measurements. The required experimental time is primarily limited by the diffusion rate (equilibration time), in this case of pyridine through zeolite pores. Thus, the time constraint is more of a challenge of liquid-phase diffusion than a technique limitation. One of the primary motivations for this technique is to provide a potential method for fundamental thermodynamic estimates in those cases when diffusion times are slow, and simpler (faster) techniques such as liquid-phase calorimetry will provide poor thermodynamic estimates due to diffusion limitations.

Regarding the choice of the probe molecule, it is true that pyridine is a molecule well-suited for ATR-FTIR studies, as it interacts strongly with zeolite micropores, and is able to differentiate Bronsted and Lewis acid sites. We stress that pyridine is far from the only molecule suitable for this spectroscopic technique, as demonstrated in Figure R1, in which alcohols with different molecular sizes are used to probe their interactions with zeolites of specific pore diameters.

Action 1: A reference to a similar technique using MAS NMR, alongside comments about the limitations of spectroscopic adsorption and in-situ reaction techniques are presented on page 20-21.

“Concerning experimental methods, in recent work from the Shanks group, high-resolution MAS NMR was used to study mixed solvent interactions with BAS at the solid-liquid interface over supported sulfonic acid materials as well as H/ZSM-5.¹³ The technique was able to distinguish bulk water from water interacting with the BAS, and used water’s chemical shift as a metric of the relative acidity over the composition range in a mixture with d₆-DMSO. Note that for any spectroscopic technique, including the ATR-FTIR method in this work, there are two potential limitations regardless of whether the technique is used for adsorption or for in-situ reaction application. The first is that the technique should be able distinguish the species of interest, e.g., pyridine vs. pyridinium in this case, or reactant vs. intermediate, and the second is that the species of interest (e.g., intermediates) should be stable enough to exist in an observable quantity. Modular excitation spectroscopy (MES) with phase-sensitive detection (PSD) can improve the surface-sensitivity of ATR-FTIR for adsorbates and surface-intermediates in many cases where these species are challenging to observe.²³⁻²⁴”

Comment 2: Other comment: In the abstract the authors should focus on the main aspects and results. In the present form, the first part is rather an introduction

Keywords: better ATR-FTIR or ATR-IR than FTIR only

Page 9: Langmuirian in shape – this is not an appropriate wording

Page 12 less enthalpically - this is also not an appropriate wording

Figure 4B: the units of K' values are missing

Page 19: DFT calculations of? . . . in solvent-saturated zeolite pores

Response 2: We appreciate the reviewer’s comment and agree with his suggestions for improving the abstract. His/her specific, page-referenced suggestions are also appreciated and amended.

Action 2: The abstract is significantly modified to read less like an introduction:

“Selecting the proper solvent is a major challenge in liquid phase catalysis, as predictive understanding of how the solvent affects reactions requires further development of quantitative experimental techniques. In this work, an attenuated total reflection infrared spectroscopy technique is developed to quantitatively measure adsorption isotherms on porous materials in solvent and decouple the thermodynamic contributions of van der Waals interactions within zeolite pore walls from those of pore-phase proton transfer. The effect of solvent identity, zeolite silicon

to aluminum ratio, hydrophobicity, and pore diameter on both confinement and subsequent protonation thermodynamics of pyridine are quantitatively examined. While both the pore diameter and the solvent identity dramatically impact the confinement (adsorption) step, the solvent identity plays a dominant role in proton-transfer. Density functional theory and ab initio molecular dynamics simulations investigate the preferred structure of a proton in the vicinity of co-adsorbed, pore-phase solvent molecules, including the formation of a protonated water-cluster, a protonated acetonitrile dimer, and zeolite framework-bound protons in the case of non-polar solvents. Both DFT and experimental methods result in increasingly favorable pore-phase proton transfer to pyridine in the order: water < acetonitrile < 1,4 – dioxane. Equilibrium methods unaffected by mass transfer limitations are outlined for quantitatively estimating fundamental thermodynamic values, including decoupled equilibrium constants for confinement (K_{conf}) and pore-phase protonation (K_{prot}), adsorption free-energies (ΔG_{ads}) from the bulk liquid into the pore-phase, zeolite pore-phase activity coefficients with co-adsorbed solvent (γ_z), and pore-phase standard-state chemical potentials ($\mu_{\text{pyr,Z}}^{\circ} - \mu_{\text{pyr,L}}^{\circ}$) using statistical thermodynamics. A method for estimating adsorption enthalpies (ΔH_{ads}) and entropies (ΔS_{ads}) is outlined using the example of pyridine adsorption into Si/ZSM-5 in liquid water, where values of -30 kJ/mol and -53 J/mol K are determined, respectively.”

References:

1. Gould, N. S. & Xu, B. Catalyst Characterization in the Presence of Solvent: Development of Liquid Phase Structure-Activity Relationships. *Chem. Sci.* **9**, 281–287 (2018).
2. Dyson, P. J. & Jessop, P. G. Solvent effects in catalysis: rational improvements of catalysts via manipulation of solvent interactions. *Catal. Sci. Technol.* **6**, 3302–3316 (2016).
3. Mellmer, M. A., Sanpitakseree, C., Demir, B., Bai, P., Ma, K., Neurock, M. & Dumesic, J. A. Solvent-enabled control of reactivity for liquid-phase reactions of biomass-derived compounds. *Nat. Catal.* **1**, 199-207 (2018).
4. Gould, N. S. & Xu, B. Effect of liquid water on acid sites of NaY: An in situ liquid phase spectroscopic study. *J. Catal.* **342**, 193–202 (2016).
5. Gould, N. S. & Xu, B. Quantification of acid site densities on zeolites in the presence of solvents via determination of extinction coefficients of adsorbed pyridine. *J. Catal.* **358**, 80–88 (2018).
6. Bai, P., Siepmann, J. I. & Deem, M. W. Adsorption of Glucose into Zeolite Beta from Aqueous Solution. *AIChE J.* **59**, 3523–3529 (2013).
7. Anastopoulos, I. & Kyzas, G. Z. Are the thermodynamic parameters correctly estimated in liquid-phase adsorption phenomena? *J. Mol. Liq.* **218**, 174-185 (2016).

8. Yang, S., Zhao, D., Zhang, H., Lu, S., Chen, L., Yu, X. Impact of environmental conditions on the sorption behavior of Pb(II) in Na-bentonite suspensions. *J. Hazard. Mater.* **183**, 632–640 (2010).
9. Sharma, S. K., *Green Chemistry for Dyes Removal from Waste Water: Research Trends and Applications*, Scrivener-Wiley, USA, (2015).
10. Gould, N. S. & Xu, B. Temperature-Programmed Desorption of Pyridine on Zeolites in the Presence of Liquid Solvents. *ACS Catal.* **8**, 8699–8708 (2018).
11. Mellmer, M. A., Sener, C., Gallo, J. M. R., Luterbacher, J. S., Alonso, D. M. & Dumesic, J. A. Solvent effects in acid-catalyzed biomass conversion reactions. *Angew. Chemie - Int. Ed.* **53**, 11872–11875 (2014).
12. Walker, T. W., Chew, A. K., Li H., Demir, B., Zhang, Z. C., Huber, G. W., Van Lehn, R. C., & Dumesic, J. A. Universal kinetic solvent effects in acid-catalyzed reactions of biomass-derived oxygenates. *Energy Environ. Sci.* **11**, 617-628 (2018).
13. Johnson, R. L., Hanrahan, M. P., Mellmer, M. A., Dumesic, J. A., Rossini, A. J. & Shanks, B. H. The Solvent-Solid Interface of Acid Catalysts Studied by High Resolution MAS NMR. *J. Phys. Chem. C.* **121**, 17226-17234 (2017).
14. Vener, M. V., Rozanska, X. & Sauer, J. Protonation of water clusters in the cavities of acidic zeolites: $(\text{H}_2\text{O})_n \cdot \text{H}$ -chabazite, $n = 1-4$. *Phys. Chem. Chem. Phys.* **11**, 1702-1712 (2009).
15. Krossner, M. & Sauer, J. Interaction of Water with Bronsted Acidic Sites of Zeolite Catalysts. Ab Initio Study of 1:1 and 2:1 Surface Complexes. *J. Phys. Chem.* **100**, 6199-6211 (1996).
16. Wang, M., Jaegers, N. R., Lee, M. S., Wan, C.; Hu, J. Z., Shi, H., Mei, D. H., Burton, S. D., Camaioni, D. M., Gutierrez, O. Y., Glezakou, V. A., Rousseau, R., Wang, Y. & Lercher, J. A. Genesis and Stability of Hydronium Ions in Zeolite Channels. *J. Am. Chem. Soc.* **141**, 3444-3455 (2019).
17. Kletnieks, P. W., Ehresmann, J. O., Nicholas, J. B., & Haw, J. F. Adsorbate Clustering and Proton Transfer in Zeolites: NMR Spectroscopy and Theory. *ChemPhysChem* **7**, 114-116 (2006).
18. Bordiga, S., Regli, L., Lamberti, C., Zecchina, A., Bjorgen, M. & Lillerud, K. P., FTIR Adsorption Studies of H_2O and CH_3OH in the Isostructural H-SSZ-13 and H-SAPO-34: Formation of H-bonded Adducts and Protonated Clusters. *J. Phys. Chem. B* **109**, 7724-7732 (2005).
19. Ben-Naim, A. *Statistical Thermodynamics for Chemists and Biochemists*. Springer US., 1992.
20. Janda, A., Vlaisavlijevich, B., Lin L-C., Sharada, S. M., Smit, B., Head-Gordon, M., & Bell, A. T. Adsorption Thermodynamics and Intrinsic Activation Parameters for Monomolecular Cracking of n-Alkanes on Brønsted Acid Sites in Zeolites. *J. Phys. Chem. C* **119**, 10427-10438 (2015).

21. Huang, B., Bai, P., Neurock, M., Davis, R. J. Conversion of n-hexane and n-dodecane over H-ZSM-5, H-Y, and Al-MCM-41 at supercritical conditions. *Appl. Catal. A*. **546**, 149-158 (2017).
22. Madon, R. J. & Iglesia, E. Catalytic reaction rates in thermodynamically non-ideal systems. *J. Mol. Catal. A Chem.* **163**, 189–204 (2000).
23. Müller, P. & Hermans, I. Applications of Modulation Excitation Spectroscopy in Heterogeneous Catalysis. *Ind. Eng. Chem. Res.* **56**, 1123–1136 (2017).
24. Urakawa, A., Bürgi, T. & Baiker, A. Sensitivity enhancement and dynamic behavior analysis by modulation excitation spectroscopy: Principle and application in heterogeneous catalysis. *Chem. Eng. Sci.* **63**, 4902-4909 (2008).

REVIEWERS' COMMENTS:

Reviewer #1 (Remarks to the Author):

The authors have sufficiently addressed my comments and I recommend that this manuscript be published.

Reviewer #2 (Remarks to the Author):

While I still have my doubts about the validity of the results such as concentration/volume determination, determining Gibbs free energies from concentrations instead of activities for a zeolite system that should be very non-ideal, and related, the ability to reach the dilute/Henry regime in the small pore zeolite systems, I think the paper can be published in its current form. It appears that the error bars are small and the paper should stimulate further research in this important area.

Reviewer #4 (Remarks to the Author):

The authors have recognized the comments and revised the manuscript accordingly.

It is recommended for publication now.